# Modelling Current and Future Forest Fire Susceptibility in north-east Germany

Katharina H. Horn[1], Stenka Vulova[1], Hanyu Li[1], and Birgit Kleinschmit[1]

[1]Geoinformation in Environmental Planning Lab, Technische Universität Berlin, Straße des 17. Juni 145, 10623 Berlin, Germany

**Correspondence:** Katharina H. Horn (k.horn@tu-berlin.de)

**Abstract.** Preventing and fighting forest fires has been a challenge worldwide in recent decades. Forest fires alter forest structure and composition, threaten people's livelihoods, and lead to economic losses, as well as soil erosion and desertification. Climate change and related drought events, paired with anthropogenic activities, have magnified the intensity and frequency of forest fires. Consequently, we analysed forest fire susceptibility (FFS), which can be understood as the likelihood of fire occurrence in a certain area. We applied Random Forest (RF) machine learning (ML) algorithm to model current and future FFS in the federal state of Brandenburg (Germany) using topographic, climatic, anthropogenic, soil, and vegetation predictors. FFS was modelled at a spatial resolution of 50 metres for current (2014-2022) and future scenarios (2081-2100). Model accuracy ranged between 69 % (RF_test) and 71 % (LOYO), showing a moderately high model reliability for predicting FFS. The model results underscore the importance of anthropogenic parameters and vegetation parameters in modelling FFS on a regional level. This study will allow forest managers and environmental planners to identify areas, which are most susceptible to forest fires, enhancing warning systems and prevention measures.

## 1  Introduction

Over the past decades, climate change has been leading to a higher intensity and frequency in extreme weather events all over the planet (Kemter et al., 2021; Silva et al., 2018; Wu et al., 2021). In Germany, very low precipitation has been occurring more frequently in the last six years, leading to an increased number of forest fires (Gnilke and Sanders, 2021). Long periods of drought have been causing soils and vegetation to dry out substantially. Especially in forests, the drying out of trees, underground vegetation, litter, and soils is making forests highly flammable (Littell et al., 2016). Consequently, it is crucial to understand the conditions that cause the emergence and spread of forest fires as well as to detect the areas that are most prone to forest fires (Ambadan et al., 2020). This way, forest fire prevention and management strategies can be improved, decreasing the subsequent potential threats to forests, population and infrastructure located in proximity to forests. In the long run, this may also decrease the financial costs of climate change (Chicas and Østergaard Nielsen, 2022).

Apart from meteorological conditions, forest fires are influenced by a number of environmental factors, including soil moisture, topography, sun exposure, lightning strikes, and wind (He et al., 2022; Saidi et al., 2021; Wang et al., 2021). Moreover, they are closely linked to human influence, encompassing the expansion of infrastructure in proximity to forests, as well as the

utilisation of forests for recreational purposes (Ghorbanzadeh et al., 2019). On a European scale, a study by El Garroussi et al. (2024) shows that 96 % of wildfires are triggered by human influence. In a similar vein, Gnilke and Sanders (2021) state that up to 50 % of the burnt area from forest fires in Germany is caused by human action. German forest fire statistics identified human negligence as the most important factor in the occurrence of forest fires (Federal Office for Agriculture and Food, 2023). Thus, anthropogenic influences should be carefully considered along with other parameters when analysing forest fires (He et al., 2022; Ruffault and Mouillot, 2017).

Forest fires and the assessment of meteorological, climatic, and anthropogenic parameters have been addressed in numerous studies. Some of them analyse the fire risk of certain regions (Ambadan et al., 2020; Saidi et al., 2021), whereas others focus on the identification of parameters influencing forest fire emergence (He et al., 2022; Ruffault and Mouillot, 2017). For example, Saidi et al. (2021) developed a GIS-remote sensing approach to investigate forest fire risk in Tunisia, whereas He et al. (2022) studied the drivers of bushfires in New South Wales, Australia over a time period of 40 years. The current state of research on forest fires suggests that topography, climate, land use, and anthropogenic influences are the most influential parameters (Abdollahi and Pradhan, 2023; Cilli et al., 2022; Ghorbanzadeh et al., 2019; He et al., 2022; Ruffault and Mouillot, 2017; Saidi et al., 2021; Li et al., 2024). For example, Ruffault and Mouillot (2017) consider human influence, land cover, and weather conditions for the assessment of influencing factors for wildfires in the French Mediterranean region.

FFS can be analysed with a variety of methodological approaches, including knowledge-based approaches, such as hierarchical weighting (Busico et al., 2019), ML and statistical approaches, or hybrid approaches (Chicas and Østergaard Nielsen, 2022). ML algorithms include RF (Cilli et al., 2022; He et al., 2022; Milanović et al., 2021; Oliveira et al., 2012, 2016), boosting models (Ruffault and Mouillot, 2017; Wang et al., 2021), and artificial neural networks (Ghorbanzadeh et al., 2019). Previous research on FFS has been focusing on bigger research areas (Busico et al., 2019; He et al., 2022; Saidi et al., 2021), whereas research on a smaller scale has fallen short. However, geodata and remote sensing data at high spatial resolution allow for detailed analysis to enhance forest fire research on a local scale. Especially regarding climate change and the growing likelihood of weather extremes such as droughts, local FFS modelling is essential for identifying key drivers on a local scale. This way, improved prevention and management strategies of forest fires can be provided. While future climate data now enables the modelling of future forest fire susceptibility (FFS), those types of studies remain scarce (Busico et al., 2019), indicating significant untapped potential for enhancing forest fire prevention efforts.

This study focuses on the analysis of forest fires in Brandenburg, Germany. Due to a high percentage of coniferous forest, this federal state has been particularly prone to forest fires in the past. Furthermore, remnants of old munitions at former military training sites have been causing forest fires in Brandenburg in 2018 and 2019 (Gnilke et al., 2022). Although this issue has been addressed by German newspapers, it has received minimal attention in scientific research (Feng et al., 2022). Therefore, this study aims to predict FFS in Brandenburg under two current (2016 and 2022) and two future scenarios (2081-2100) using geodata and remote sensing data at high spatial resolution and the Random Forest (RF) machine learning (ML) algorithm. Following Zhang et al. (2019), FFS in this study represents "the probability estimation of fire occurrence". In addition to topographic, vegetation, and soil parameters, this study incorporates a comprehensive set of anthropogenic and land use parameters, including new predictors such as the distance to campsites and military training sites, to expand existing research on

forest fires. To our knowledge, only few studies have analysed FFS at a high spatial resolution so far (Ghorbanzadeh et al., 2019; Suryabhagavan et al., 2016; Razavi-Termeh et al., 2020; Pourtaghi et al., 2015) and we do not know of any studies that modelled future FFS at a high spatial resolution. Within the scope of this investigation, the following research questions will be answered:

    a) Which variables are most significant in terms of forest fire spread in north-east Germany?

    b) Which areas in Brandenburg are most susceptible to forest fires now? How will these areas change considering future climate conditions?

## 2    Materials and Methods

### 2.1    Study area

The federal state of Brandenburg (Fig. 1) was selected as the study area for modelling FFS under current and future scenarios. Brandenburg is located in the north-east of Germany. With sandy or sandy-loamy soils and a high number of rivers and lakes, the federal state is characterised by a periglacial landscape. Agriculture and managed forests are the main land uses. The forests are dominated by pine trees (Pinus sylvestris L.) (Matos et al., 2010) and the climate is characterised by rather dry summer months. The combination of these conditions is linked to a medium to high forest fire risk (Holsten et al., 2009; Matos et al., 2010; Reyer et al., 2012; Thonicke and Cramer, 2006). Comparing all German federal states, Brandenburg is the federal state that has been most affected by forest fires (Gnilke and Sanders, 2021), which is why it was selected for this study.

### 2.2    Current and future forest fire susceptibility scenarios

The aim of this research is to compare FFS under different temporal scenarios. To do so, current and future FFS in the federal state of Brandenburg were modelled. To represent the current state, the years of 2016 and 2022 were selected after carefully analysing the monthly precipitation sums and mean monthly air temperature of Brandenburg between 2014 to 2022 (see Fig. S 1 and S 2 in the Supplement). Based on this analysis, 2016 was characterised by average climatic conditions, whereas 2022 was characterised by conditions of drought (low precipitation rates). Consequently, the scenario of 2016 was considered as a baseline scenario with average climatic conditions. In contrast to 2016, the scenario of 2022 represents a very dry year, which can be expected to occur more frequently due to the expected increase in extreme weather events in the future (Silva et al., 2018; Wu et al., 2021).

The future scenarios of FFS cover the period of 2081 to 2100 using the socio-economic pathway (SSP) 5-8.5. SSPs are different projections of future greenhouse gas emissions under distinct potential political and socioeconomic developments. The SSPs range from SSP1-1.9 to SSP5-8.5, covering $CO_2$ concentrations ranging from 393 to 1135 ppm until 2100. SSP5-8.5 represents "a high fossil-fuel development world throughout the 21st century" (Meinshausen et al., 2020). We decided to use SSP5-8.5 from the Global Climate Model (GCM) MPI-ESM-1-2-HR.Xu et al. (2023) state that this GCM reflects future drought conditions rather well, which is why it was selected for this study. The climate data (monthly average minimum temperature

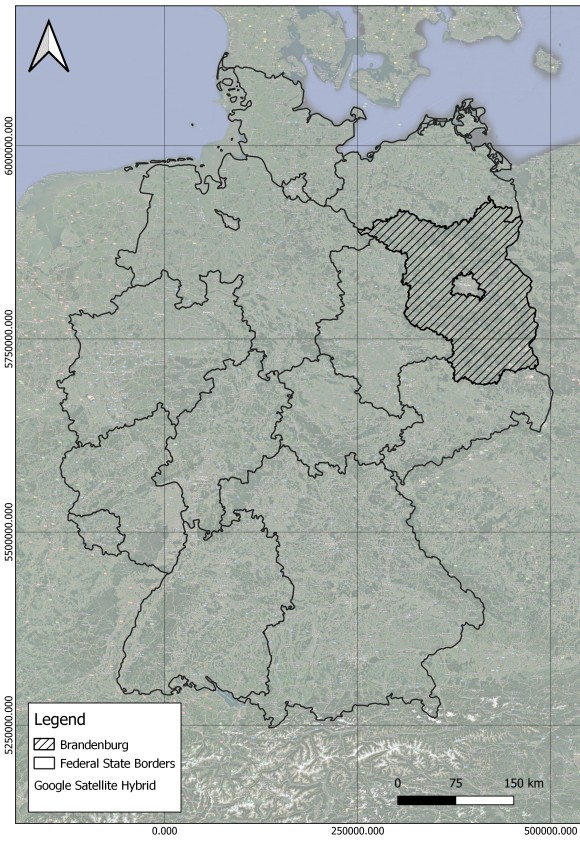

**Figure 1.** The federal state of Brandenburg in north-east Germany. Basemap © 2024 TerraMetrics, Google, GeoBasis-DE/BKG (© 2009). Border layers © BKG (2024) dl-de/by-2-0 (Data not changed).

(°C), monthly average maximum temperature (°C), and monthly total precipitation (mm)) were downloaded from WorldClim (www.worldclim.org). This website provides gridded multi-annual data sets based on different GCMs for different socio-economic pathways (SSPs) and different time periods between 2021 to 2100 up to 30 arc seconds (~1 km) spatial resolution. In order to include future land cover changes into the future predictions, future FFS was predicted twice: a) including only

95 projected meteorological data for 2081-2100; and b) including projected meteorological data for 2081-2100 and projected land cover data. Within the Fig. 2, 4, 5, 6, and 7, as well as in Table 3 the latter will be labeled with "*". Additionally, a third future scenario based on the SSP3-7.0 was predicted. The results can be found in the Supplement (Fig. S 10 to S 13). After analysing the monthly frequency of forest fires in the federal state of Brandenburg, the month of June was selected for the prediction of the four scenarios, since forest fire data showed the highest number of forest fires in this month between 2014 to 2022 (Lower

Forestry Authority of the State of Brandenburg, 2023). For model training, we used all available forest fire events of all months between 2014 to 2022 and pre-processed climatic data sets in accordance with the available forest fire data.

## 2.3 Data

### 2.3.1 Forest fire data

To model FFS in Brandenburg under different scenarios, forest fire data as well as a set of predictor datasets were acquired and pre-processed. Data including statistical and geospatial information on forest fires in Brandenburg were provided by the Lower Forestry Authority of the State of Brandenburg (2023), an institution that focuses on analysing the vitality of forests in the federal state (Lower Forestry Authority of the State of Brandenburg, 2023; Ministry for Rural Development, Environment and Agriculture in Brandenburg, 2023). The Lower Forestry Authority of the State of Brandenburg (2023) provided data containing the following information: forest district number, section, date and hour, cause of fire, burnt area (ha), and X-Y coordinates of the fire ignition point.

### 2.3.2 Predictor variables

To model FFS in Brandenburg, a set of 20 predictors were selected for the analysis. The predictor variables are shown in Table 1 (also see Fig. S 4 in the Supplement). They cover meteorology, vegetation, topography, soil, anthropogenic influences and land use and land cover (LULC) and were identified as most relevant for modelling FFS based on an extensive literature review. In the following sections, the predictor variables will be presented in more detail.

a) Meteorology

To assess climatic conditions for both the current and future scenarios, air temperature and precipitation were selected. Since climate change and the consequent increase in extreme weather events such as meteorological droughts around the world may increase the frequency and intensity of forest fires in the future (Abdollahi and Pradhan, 2023; Silva et al., 2018), air temperature and precipitation patterns are crucial for the analysis of FFS. Further climatic parameters such as wind speed, solar radiation or lightning strikes may impact the emergence of forest fires as well (Abdollahi and Pradhan, 2023; Busico et al., 2019). However, for the scope of this work the focus remained on air temperature and precipitation, since both current and projected data was only available for those climatic parameters. Following the suggestions by He et al. (2022), we used monthly climate data between 2013 to 2022, which was aggregated to three months to incorporate precipitation and air temperature prior to the occurrence of a forest fire. Several forest fire related studies have used a monthly aggregation of meteorological data sets to model forest fires (Busico et al., 2019; Wang et al., 2021; He et al., 2022). He et al. (2022) further argue that future studies should consider a monthly or quarterly aggregation of meteorological data when investigating forest fires. Especially in order to identify conditions of meteorological droughts prior to the emergence of a forest fire, we followed the methodology of other authors that used a three-month aggregation of the broadly used SPEI drought index to identify meteorological droughts (Zhou et al., 2023; Wen et al., 2020; Guo et al., 2018).

b) Vegetation

The type and condition of vegetation is a crucial factor in the emergence of forest fires (Abdollahi and Pradhan, 2023). Several studies have shown that monocultural forests are more likely to be affected by forest fires not only in number, but also in extent (Afreen et al., 2011; Bauhus et al., 2017). For example, Bauhus et al. (2017) state that coniferous species such as pine trees tend to be highly flammable, which is mainly caused by their resins and oils. Furthermore, the distance to the forest edge can impact tree vitality and the consequent vulnerability to droughts (Buras et al., 2018). Buras et al. (2018) analysed the tree mortality of Scots pine forests by comparing trees on the forest edge and trees in the interior of the forests. Their results show an increase in vulnerability to drought of trees located at forest edges, resulting in higher mortality and decreased vitality. Consequently, the selected vegetation-related predictors were the percentage of broadleaf forest, canopy height, tree cover density, and the distance to forest edges.

c) Topography

Numerous studies have shown the influence of topography on the emergence of forest fires, which is why topographic parameters are commonly used for studying forest fires (Abdollahi and Pradhan, 2023; Busico et al., 2019; Ghorbanzadeh et al., 2019; He et al., 2022; Maingi and Henry, 2007; Saidi et al., 2021; Wang et al., 2021). For example, Preston et al. (2009) have pointed out that bushfires spread with higher velocity and intensity on upward slopes. Furthermore, they discuss how aspect impacts sun and wind regimes, which may influence forest fires as well. In this regard, Busico et al. (2019) conclude that northern aspects decrease the likelihood of forest fire ignition. Besides slope and aspect, elevation has been pointed out as a significant parameter for forest fires (He et al., 2022; Maingi and Henry, 2007). Chicas and Østergaard Nielsen (2022) performed an extensive analysis of existing studies on mapping FFS, confirming that slope, elevation, aspect and topographic wetness index (TWI) are the most commonly used topographic parameters. Following their assessment, those four parameters were selected for the scope of this study.

d) Soil

The spread of forest fires is greatly influenced by the characteristics of the soil and its moisture content (He et al., 2022). Therefore, it was considered important to include different soil characteristics as predictor variables. The soil depth chosen for the soil predictors was 0-5 cm, since fires are usually initiated on the soil surface (Badía-Villas et al., 2014; Mallik et al., 1984). The water retention capacity of soils is significantly influenced by their structure, such as the relative proportions of sand and silt. Soil types characterised by larger pore sizes, such as sandy soils, typically exhibit low water retention capabilities, leading to arid conditions and a diminished field capacity. Conversely, soils with intermediate pore sizes, or silty soils, have higher moisture levels and more water available for plants (Amelung et al., 2018). Therefore, the proportion of sand particles ($> 0.05$ mm) in the fine earth fraction (sand) and the proportion of silt particles ($\geq 0.002$ mm and $\leq 0.05$ mm) in the fine earth fraction (silt) were selected for the analysis. Similarly, both bulk density of the fine earth fraction (bdod) and organic carbon density (ocs) can serve as proxies for water retention and therefore for the flammability of the soil (Oyonarte et al., 1998). For example, Oyonarte et al. (1998) have shown a high correlation between water retention and organic carbon, as well as bulk density, which underlines their potential influence on FFS. Thus, bulk density of the fine earth fraction and organic carbon density were used as predictor variables as well.

e) Anthropogenic influences & land use and land cover (LULC)

Finally, anthropogenic factors as well as LULC have been shown to influence the emergence of past forest fires in Brandenburg (Gnilke and Sanders, 2021). The data provided by the Lower Forestry Authority of the State of Brandenburg (2023) on causes of forest fire ignitions in Brandenburg between 2014 to 2022 (see Table S 2 in the Supplement) confirms this statement. In a similar vein, He et al. (2022) argue that human activities such as the construction of transportation networks and other types of infrastructure influence forest fire emergence on a local scale. Therefore, they highly recommend including anthropogenic factors into the analysis of forest fires. Likewise, Ghorbanzadeh et al. (2019) relate the increase in forest fires not only to the changing climate, but also to anthropogenic aspects such as human activities or demographic expansion. Thus, to predict FFS in northern Iran, they included proximity to villages, streets, and recreational areas, as well as aspects of land use as predictor variables. The latter has been emphasised by Busico et al. (2019) as well, who stated that anthropogenic land use significantly contributes to forest fire emergence. Consequently, to include anthropogenic influences as well as aspects of LULC, distance to urban settlements, streets, railways, campsites, water bodies and military sites were selected as predictor variables. According to the respective data set, we understand "distance to urban settlements" as the distance to any type of constructed above-ground building (European Environment Agency [EEA], 2020b). We assume that this predictor can show (ir-) regular human presence at these places that may be related to an increased FFS. Furthermore, to address future land cover changes, we included a data set on projected land cover change in 2050 provided by Esri Environment (2021). To our knowledge, this was the only available data set with a high spatial resolution to show future land cover changes, which is why it was selected for this study. Table 1 provides an overview of the predictors as well as their characteristics and origin.

## 2.4  Data processing

RStudio version '2023.12.0.369' with R version 4.3.1 (2023-06-16 ucrt) was used for data pre-processing, analysis, RF modelling and computation of statistics, graphs and maps. Geospatial packages such as terra, sf, maptools and ggplot2 were used for data pre-processing and analysis. The caret package was used for modelling and the computation of performance metrics. The dplyr and readxl packages were used for the analysis and formatting of the forest fire data. The open source software QGIS 3.28.10-Firenze was used for processing, analysis, and visualisation of the geodata. Figure 2 provides an overview of the main data processing steps that will be explained in the following sections.

a) Pre-processing of predictor layers

Prior to modelling FFS under current and future scenarios, the necessary datasets were downloaded and pre-processed. Pre-processing steps involved projecting the data to the same coordinate reference system (EPSG 25833), cropping to the geographic extent of Brandenburg, masking the forest areas in Brandenburg, and resampling to a spatial resolution of 50 metres using bilinear interpolation for numeric variables, and nearest neighbour interpolation for factor variables. Furthermore, several predictor datasets such as distance to campsites or military areas were created based on available data from OpenStreetMap Contributors (2023) or LGB State Office for Land Surveying and Geoinformation Brandenburg (2023). The topographic predictors slope, aspect and TWI were computed in RStudio based on the digital elevation model derived from LGB State Office

**Table 1.** Predictor variables for modelling forest fire susceptibility in Brandenburg.

| Category | Predictor | Abbreviation | Data Source | Spatial Resolution | Temporal Resolution | Unit |
|---|---|---|---|---|---|---|
| Meteorology | Air temperature (current scenario) | airtemp | GWS Climate Data Center (2023b) | 1 km | monthly mean | 1/10 °C |
| | Air temperature (future scenario) | airtemp | Fick and Hijmans (2022) | 30 arcsec | multi-annual monthly mean | °C |
| | Precipitation (current scenario) | precip | GWS Climate Data Center (2023a) | 1 km | monthly sum | mm |
| | Precipitation (future scenario) | precip | Fick and Hijmans (2022) | 30 arcsec | multi-annual monthly sum | mm |
| Vegetation | Tree cover density | tcd | EEA (2020c) | 10 m | - | % |
| | Distance to forest edge | forestedge | EEA (2020c) | 10 m | - | m |
| | Percentage of broadleaf forest | broadleaf | EEA (2020a) | 20 m | - | % |
| | Canopy height | canopy | Lang et al. (2023) | 10 m | - | m |
| Topography | Slope | slope | LGB State Office for Land Surveying and Geoinformation Brandenburg (2023) | 10 m | - | ° |
| | Aspect | aspect | | 10 m | - | ° |
| | Elevation | dem | | 10 m | - | m |
| | Topographic Wetness Index | twi | | 10 m | - | - |
| Soil | Bulk density of the fine earth fraction | bdod | | 250 m | - | cg/cm$^3$ |
| | Organic carbon density | ocs | | 250 m | - | hg/m$^3$ |
| | Proportion of sand particles (> 0.05 mm) in the fine earth | sand | Poggio et al. (2021) | 250 m | - | g/kg |
| | Proportion of silt particles ($\geq$ 0.002 mm and < 0.05 mm) in the fine earth | silt | | 250 m | - | g/kg |
| Anthropogenic parameters and LULC | Distance to urban settlements | urban | EEA (2020b) | 10 m | - | m |
| | Distance to streets | streets | | - | - | m |
| | Distance to railways | railways | OpenStreetMap Contributors (2023) | - | - | m |
| | Distance to campsites | campsites | | - | - | m |
| | Distance to military sites | military | | - | - | m |
| | Distance to water bodies | water | | - | - | m |
| | Distance to urban settlements (2050) | urban_2050 | Esri Environment (2021) | 300 m | - | m |

The topographic predictors slope, aspect, and TWI were computed based on the digital elevation model that was derived from LGB State Office for Land Surveying and Geoinformation Brandenburg (2023).

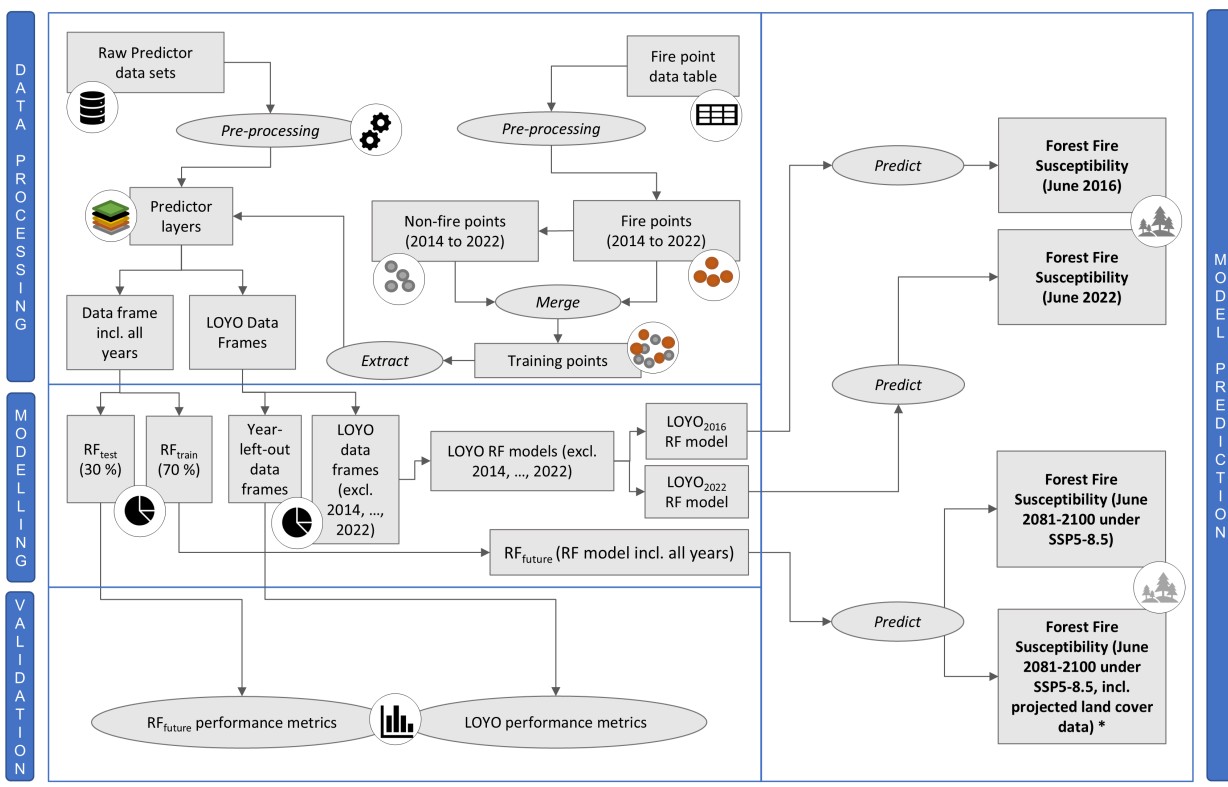

**Figure 2.** Methodological approach for modelling forest fire susceptibility under different scenarios.

for Land Surveying and Geoinformation Brandenburg (2023). A forest mask was generated by filtering all pixels with tree cover density greater than or equal to 50 % from the tree cover density dataset. Proximity rasters were computed for various features, including urban settlements, roads, railways, military sites, campsites, water bodies, and forest edges, by applying the "Proximity (raster distance)" tool in QGIS derived from the GDAL toolbox.

b)  Processing of training points

The forest fire data table provided by the Lower Forestry Authority of the State of Brandenburg (2023) served as the baseline for the creation of the training points for the RF models. Rows containing NA values were removed and the fire data points were converted to shapefile format for further processing. Looking at the statistics of the burnt area (ha) of each of the fires in Brandenburg between 2014 to 2022, the maximum burnt area of a forest fire was 422 ha. In contrast, the median burnt area was only 0.05 ha, indicating a high number of small fires and a relatively low number of big fires (see Table S 1 in the Supplement). Since the spread extent of the fires was not included in the data provided by the Lower Forestry Authority of the State of Brandenburg (2023), a circular fire spread was assumed. The diameter of a circular burnt forest fire based on the median burnt area (0.05 ha or 500 m$^2$) is ~25 m. Considering that the direction of the fire spread was unknown as well, the

doubled diameter of a median sized forest fire in Brandenburg (50 m) was assumed as a baseline for converting the forest fire points into a raster dataset (see Fig. S 3 in the Supplement). Consequently, the fire points were resampled to a raster grid with 50 m spatial resolution considering the potential fire spread in different directions. Accordingly, all the predictor variables were resampled to the same spatial resolution.

In addition to the provided set of fire points, a set of non-fire points was created that included the identical number of points per year as the pre-processed fire points from the data table provided by the Lower Forestry Authority of the State of Brandenburg (2023). To create those non-fire points, the maximum extent of each forest fire for each year was computed to identify areas where no fires occurred for each year. To do so, the fire point data table was first subsetted by year and then burnt area was estimated based on the previously described approach. The results were nine raster layers for each year between 2014 to 2022 that contained the maximum extent that was potentially burnt in that respective year. For each year, potential burnt areas were then removed from the forest mask layer to derive areas where no fires occurred. Based on the forest masks that excluded potentially burnt areas, random non-fire points were created for each year, matching the number of fires that occurred in the respective year. To do so, the randomPoints() function from the R package raptr was used.

Finally, the resulting non-fire points were merged with the fire points to complement the training points. To do so, the training points were assigned to the classes of "fire" and "non-fire", respectively. Each fire registered by the Lower Forestry Authority of the State of Brandenburg (2023) was paired with a non-fire point with the same date. To prepare the dataframe for the RF models, the training points were used to extract the geospatial information of the predictor variables using extract() function from the terra R package. The resulting data table included the spatial coordinates of all non-fire and fire points and the information of all the predictor variables at those locations. This dataframe served as the basis for training RF models to predict FFS under current and future scenarios.

## 2.5 Correlation analysis and Random Forest modelling

To assess FFS in Brandenburg under different temporal scenarios, RF classification ML algorithm was used. Precisely, a total of ten RF models were run using binary classes (fire and non-fire) for predicting current and future FFS. RF is a well known and often used ML algorithm in forestry and remote sensing (Gislason et al., 2006). In the field of forest fire research, RF has been frequently applied, achieving high accuracies (Eslami et al., 2021; He et al., 2022; Lizundia-Loiola et al., 2020; Milanović et al., 2021; Oliveira et al., 2016). The RF algorithm is based on the bagging approach, developed by Breiman (1999). It involves the growth of a set of random decision trees to form what is known as a "Random Forest" (Breiman, 2001; Kuhn and Johnson, 2013). As mentioned before, FFS is defined in this study as the estimated likelihood of a forest fire event (Zhang et al., 2019). The probability score of a pixel being predicted as a fire pixel represents its susceptibility to a forest fire.

*Model for future scenarios*

First, a model ($RF_{future}$) containing data from all the available years (2014 to 2022) was set up for the prediction of future FFS scenarios. Following Nguyen et al. (2021), the input data for modelling FFS was split into 70 % for model training ($RF_{train}$) and 30 % for testing the model performance ($RF_{test}$). We refer to the 30 % left out data as the testing dataset. Before running a RF

model, a set of tuning parameters can be set. After initially running the model, the results showed the best model performance at mtry = 2. Consequently, the model was run with mtry set to 2.

*Models for current scenarios*

For current FFS scenarios, a so-called "leave-one-year-out" (LOYO) approach was implemented in order to evaluate the models' capacity for temporal extrapolation. Leaving one year out of training and using the excluded year for testing can be used to assess how models will perform on an unseen (or future) year. In this case, the approach was used for modelling current FFS for the scenarios of 2016 and 2022. LOYO models were computed for all nine available years (2014 to 2022). For instance,

$LOYO_{2016}$ refers to a model trained on all years except 2016, which was used to predict FFS in 2016. As mentioned before, mtry was set to 2 to be consistent with the model for the future FFS scenarios.

*Performance metrics*

After training the RF models, performance metrics were calculated using the caret and rPROC packages. The confusionMa-

260 trix() function provides information on the different performance metrics such as accuracy, kappa, sensitivity, or specificity. Additionally, F1-score and AUC were computed using the rPROC package in RStudio. AUC was calculated by first computing the receiver operator characteristic (ROC) curve using the roc() function. The formulas for calculating the different performance metrics can be found in the Supplement (Table S 3). They typically range between 0 and 1, with values close to 1 implying a high model performance.

## 3 Results

### 3.1 Model accuracy

To assess the reliability of the $RF_{future}$ model in predicting FFS in Brandenburg, performance metrics and a confusion matrix (see Table S 4 in the Supplement) were computed. The training ($RF_{train}$) and testing set ($RF_{test}$) for the $RF_{future}$ model consisted of 3243 and 1388 points respectively. 487 out of 681 fire points and 520 out of 707 non-fire points were correctly classified.

The performance metrics (Table 2) for both $RF_{test}$ and the LOYO cross validation all range between 0.654 and 0.718 (excluding the kappa values), showing a moderately high model reliability of predicting FFS in Brandenburg. $RF_{test}$ had an accuracy of 0.718, reflecting the number of samples that were correctly classified as fire points. The LOYO cross validation indicates a marginally lower mean accuracy of 0.695. The precision values of LOYO cross validation (0.702) and $RF_{test}$ (0.712) illustrate the proportion of correctly assigned fire points out of all samples that were classified as fire. To further assess the performance

of the RF FFS classification, the ROC curve was computed. The area under the ROC curve (AUC) refers to the likelihood that a fire point was correctly classified (Bradley, 1997). Here, AUC is 0.694 for the LOYO cross validation and at 0.718 for $RF_{test}$. Finally, recall and F1-score metrics show similar values, indicating a moderately high model reliability. A detailed overview of all the performance metrics for every LOYO model can be found in Table S 5 in the Supplement.

**Table 2.** Overview of the validation metrics.

|  | Accuracy | Kappa | Precision | Recall | F1-Score | AUC |
|---|---|---|---|---|---|---|
| $RF_{test}$ | 0.718 | 0.435 | 0.712 | 0.714 | 0.713 | 0.718 |
| LOYO cross-validation | 0.695 | 0.388 | 0.702 | 0.654 | 0.676 | 0.694 |

## 3.2 Importance of predictor variables

Overall, distance to urban settlements, the percentage of broadleaf forest, and the distance to railways were the three most significant predictors for the $RF_{future}$ model. The importance of these predictors, as well as others, is shown in Fig. 3. Land use and anthropogenic predictors exhibited moderate to high influence for the model, such as the distance to urban settlements (100 %), the distance to railways (84.3 %), or the distance to campsites (50.9 %). Similarly, vegetation predictors showed varying degrees of influence, ranging from moderate (e.g., distance to forest edge) to high parameter importance, notably the
percentage of broadleaf forest (87.8 %). Soil predictors demonstrated medium importance, ranging from 39.9 % for organic carbon density to 53.4 % for silt content. Topographic predictors displayed varied importance, with elevation at 49.1 % and the TWI at 11.6 %. In contrast, climatic variables had a relatively minor influence on model performance, with air temperature contributing only 14.4 % and precipitation accounting for a mere 3.1 %. The value distributions of the three most significant predictors are depicted in Fig. S 5 of the Supplement. A Wilcoxon test was conducted to test significance. The notably low
p-values of the Wilcoxon tests, for example p = 5.70e-20 for the percentage of broadleaf, confirm that the value distributions of all three predictors significantly differ between fire and non-fire points. A comprehensive overview of the p-values for all predictor variables is provided in Table S 6 in the Supplement.

The value distributions of the three most significant predictors (Fig. S 5) lead to several conclusions. First, fire points tend to be closer (mean ~578 m) to urban settlements than non-fire points (mean ~813 m). Second, the distribution in the percentage
of broadleaf is ranging mainly from 0 to almost 40 % for non-fire points, whereas the percentage of broadleaf for fire points is close to 0 (excluding some outliers). Third, similarly to the distance to urban settlements, non-fire points tend to be further away from railways than fire points. To more deeply explore the relationship between key variables and FFS, partial dependence plots were created (see Fig. S 7 to S 9 in the Supplement).

## 3.3 Forest fire susceptibility under current and future scenarios

Figure 4 shows the FFS in Brandenburg for the two current scenarios, June 2016 and June 2022, as well as for the two future scenarios, June 2081-2100 under SSP5-8.5 and June 2081-2100 under SSP5-8.5 including projected land cover data. For comparison, the FFS for June 2081-2100 under SSP3-7.0 can be found in the Supplement (Fig. S 10). The values range from 0 to 100 %, reflecting the likelihood of fire ignition at each pixel (the FFS). In all four scenarios, the FFS is higher in the southern part of Brandenburg. Especially in the south of Berlin, several patches with a FFS of >= 75 % can be identified. In the north
and north-east of Brandenburg however, the FFS is rather low in all the scenarios, ranging between 0 to 20 %.

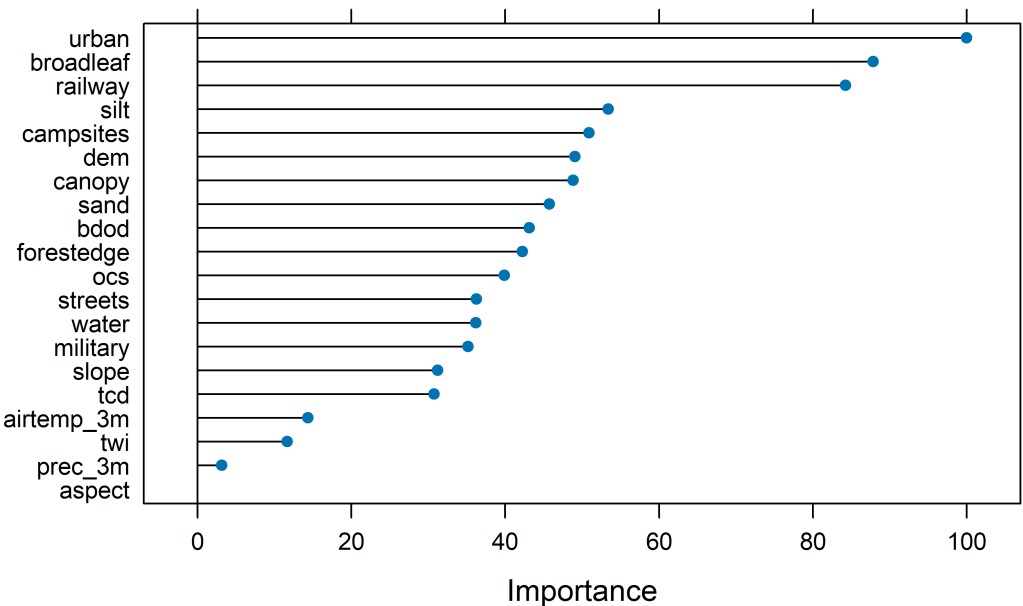

**Figure 3.** Variable importance based on the RF$_{future}$ model.

Figure 5 illustrates the anomalies in FFS relative to the reference scenario of June 2016. In the June 2022 scenario (scenario a), FFS exhibits notable positive anomalies across various regions of the federal state, with anomalies ranging from + 5 to + 15 % compared to June 2016. Many areas across Brandenburg maintain similar FFS levels as the 2022 scenario. Only a few selected small regions in the south-east and south-west exhibit negative FFS anomalies compared to June 2016. Regarding future FFS anomalies relative to June 2016, the future scenarios differ rather substantially from one another. Whereas the scenario neglecting land cover changes (b) shows positive FFS anomalies up to 15 % and more in southern, eastern, and western parts of Berlin, one area in the south shows negative FFS anomalies up to - 20 %. In comparison to the scenario based on only climatological projections, the scenario incorporating land cover changes (c) shows mostly negative FFS anomalies ranging from 0 to - 20 %, especially in the southern part of Brandenburg. The northern part of Brandenburg however is characterised by an increase in FFS in many areas, reaching anomalies up to + 20 %. Additionally, some areas in the South and West also show positive FFS anomalies. For comparison, the FFS anomalies for 2081-2100 under SSP3-7.0 can be found in the Supplement (Fig. S 11 to S 13).

Table 3 presents summary statistics of the FFS for the four scenarios. Upon comparing the values across all scenarios, it is evident that the 2016 scenario exhibits the lowest minimum value among the four. Conversely, the 2022 scenario demonstrates higher maximum and mean FFS values, suggesting a greater susceptibility compared to 2016. Notably, the mean susceptibility value for 2022 (0.419) is the highest among the four scenarios indicating the highest mean FFS. The future scenario excluding

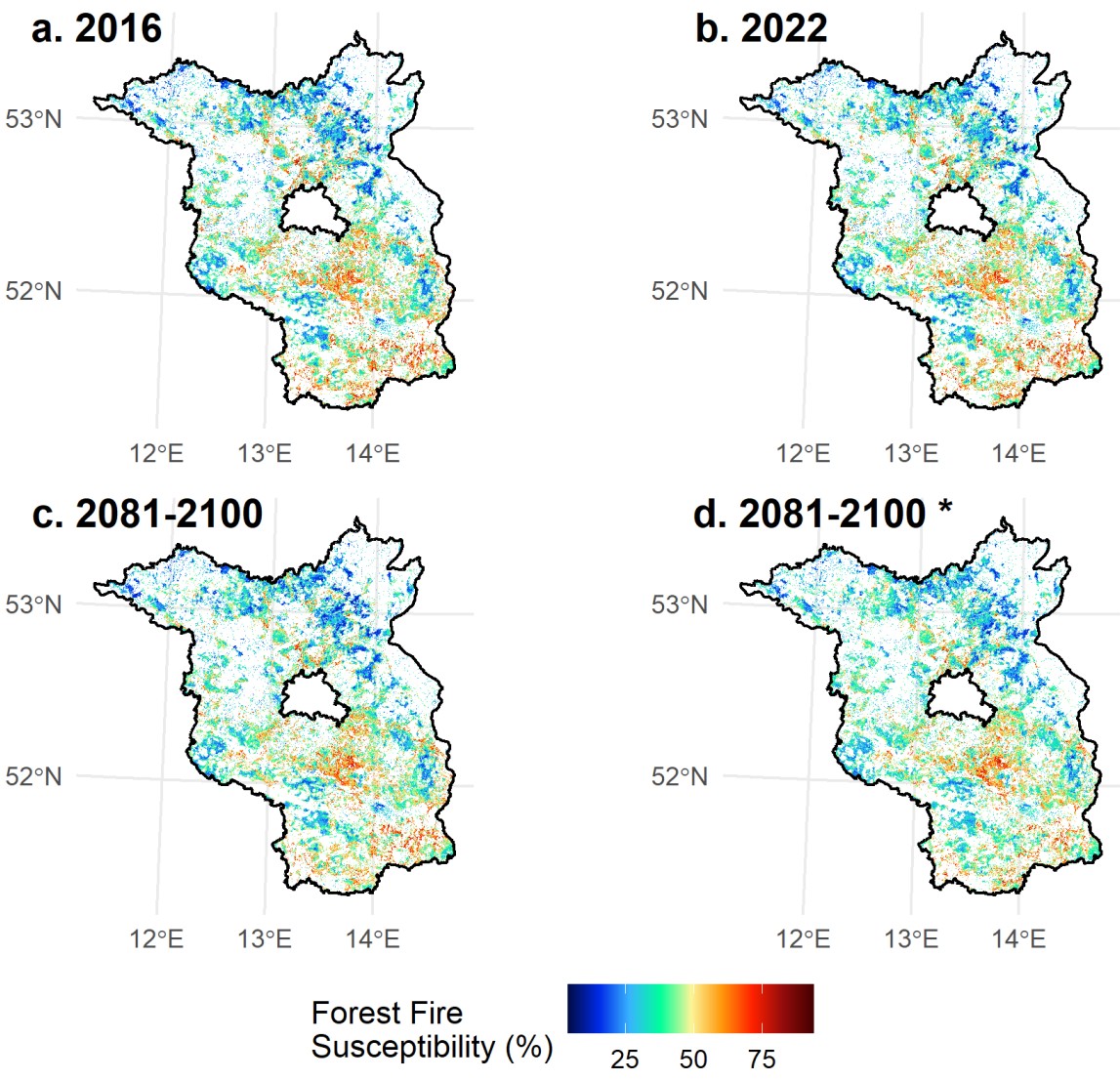

**Figure 4.** Forest fire susceptibility in Brandenburg under different scenarios. Scenarios c and d both show predicted FFS in June 2081-2100 under SSP5-8.5. Scenario d includes projected land cover data, whereas scenario c does not. Border layer © 2018-2022 GADM.

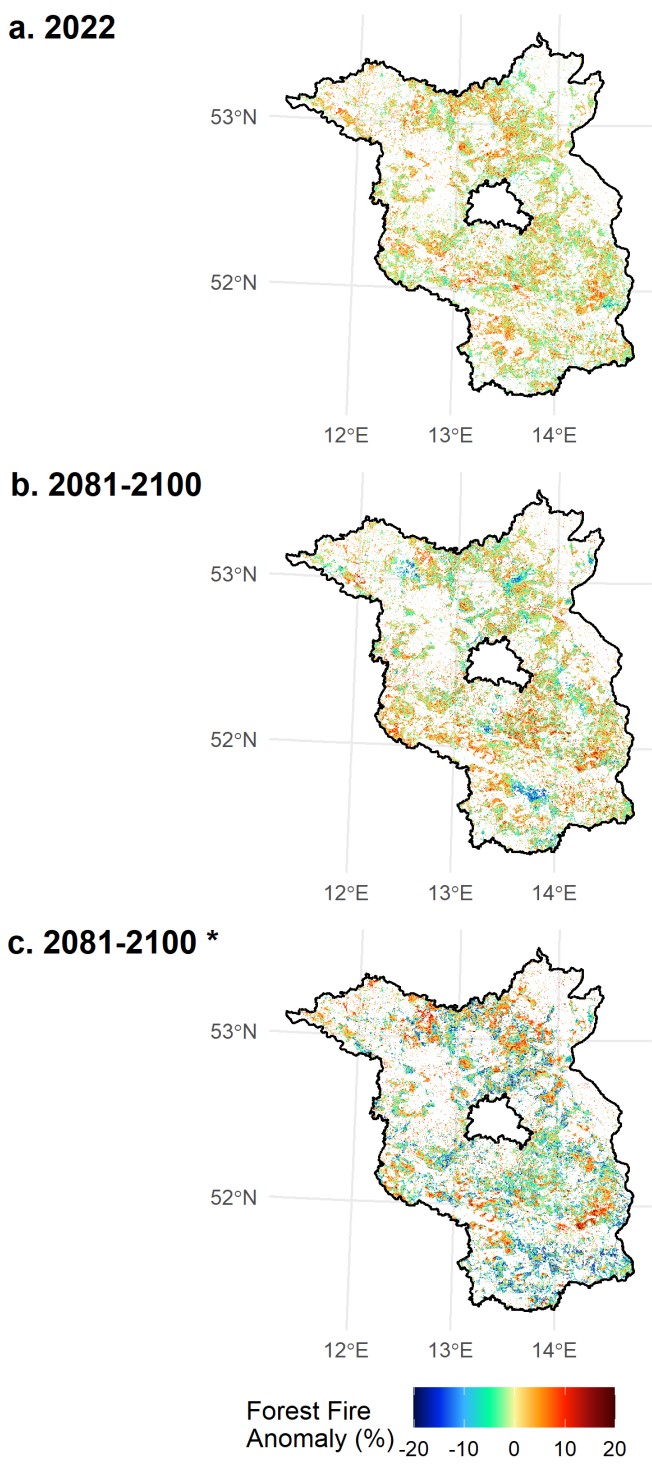

**Figure 5.** Forest fire anomalies compared to 2016. Scenarios b and c both show predicted FFS anomalies in June 2081-2100 under SSP5-8.5. Scenario d includes projected land cover data, whereas scenario c does not. Border layer © 2018-2022 GADM.

**Table 3.** Statistical overview of the four forest fire susceptibility scenarios. Scenarios "2081-2100" and "2081-2100 *" both show predicted FFS in June 2081-2100 under SSP5-8.5. Scenario "2081-2100 *" includes projected land cover data, whereas scenario "2081-2100" does not.

|  | 2016 | 2022 | 2081-2100 | 2081-2100 * |
|---|---|---|---|---|
| Minimum | 0.040 | 0.040 | 0.042 | 0.072 |
| Maximum | 0.936 | 0.964 | 0.976 | 0.878 |
| Mean | 0.409 | 0.419 | 0.417 | 0.393 |
| Standard Deviation | 0.147 | 0.146 | 0.144 | 0.116 |

projected land cover data shows the highest maximum value and only a slightly lower mean value (0.414) than the scenario of June 2022. Finally the future scenario including land cover data (*) shows the lowest maximum, mean and standard deviation FFS values compared to the other scenarios.

To assess variabilities in FFS on a local scale, a detailed zoom to an area in the west of Brandenburg is shown in Fig. 6. The four maps show the municipality of Medewitz in the west of Brandenburg. The 2016 scenario shows a fairly low FFS (Fig. 6a). The three other maps show FFS anomalies compared to 2016 (Fig. 6b to d). Whereas the scenario of 2022 shows positive anomaly values of 10 to 15 %, anomaly values are even higher in the future scenario excluding projected land cover data, reaching + 20 %. In contrast, the scenario including land cover changes (d) shows negative anomalies up to - 15 %. However, pixels in the east and south of the map show positive FFS anomalies as well.

The four zoomed-in maps in Fig. 7 depict the Crinitz municipality located in the south of Brandenburg. Whereas the June 2022 scenario (b) mainly shows anomalies close to 0, except for some pixels reaching up to + 16 %, the future scenario relying only on climatic projections (c) shows substantial negative anomalies reaching up to -20 %. Similarly, the scenario including projected land cover data (d) shows a substantial proportion of pixels with negative FFS anomalies. However, some areas in the north and southwest of the city show positive FFS anomalies.

Figures 6 and 7 show that despite the trend of overall increase in FFS between 2016 and the future scenario of 2081-2100 excluding projected land cover data (Fig. 4 and 5), FFS differs significantly across the federal state. Furthermore, the future scenario incorporating land cover changes shows substantial differences to the scenario only relying on climatic projections.

## 4 Discussion

### 4.1 The drivers of forest fire susceptibility

Overall, the climatic variables did not have a significant influence on the model performance. In contrast, the anthropogenic, LULC, and vegetation predictors showed a higher importance. The results reflect that climatic parameters do not appear to play a pivotal role for FFS (see Fig. S 6 in the Supplement). The reason for this finding may be the extent of the study area, as meteo-

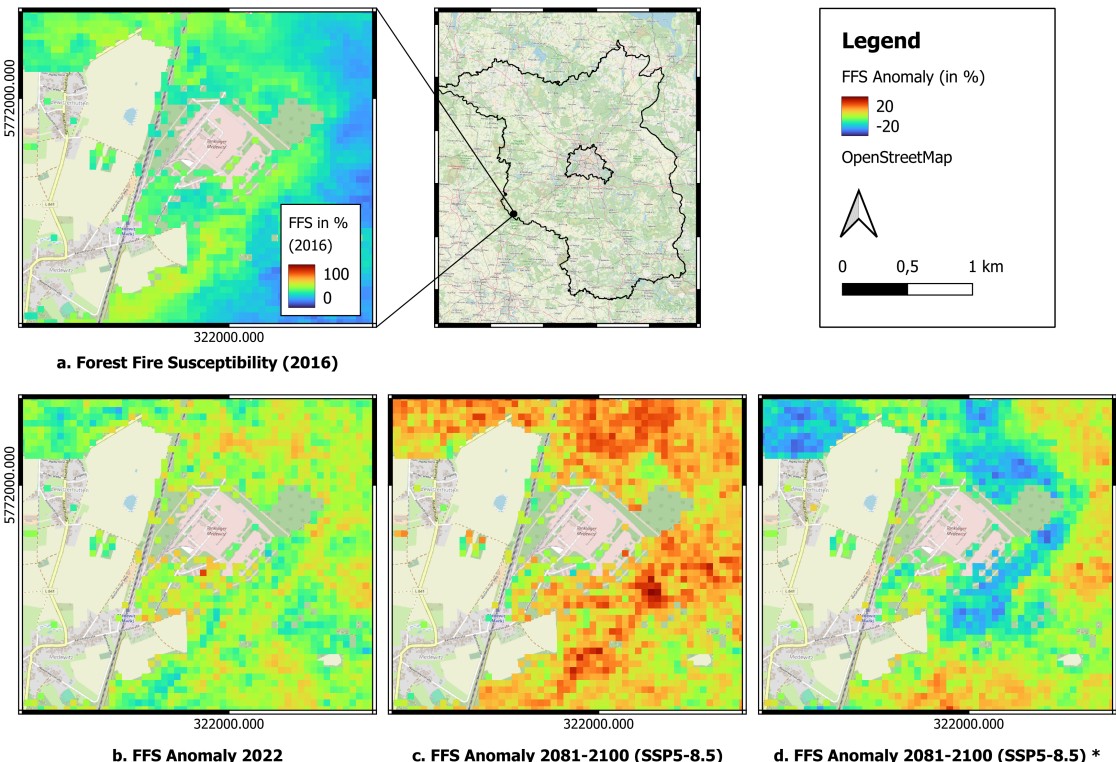

**Figure 6.** Detailed maps of FFS anomalies in the municipality of Medewitz (Brandenburg). Scenarios c and d both show predicted FFS in June 2081-2100 under SSP5-8.5. Scenario d includes projected land cover data, whereas scenario c does not. Base Map © OpenStreetMap contributors 2024. Distributed under the Open Data Commons Open Database License (ODbL) v1.0, Border layer © 2018-2022 GADM.

rological conditions do not show high spatial variation within Brandenburg. Meteorological conditions may be more important
when analysing FFS on a national or international scale (Busico et al., 2019; He et al., 2022; Li et al., 2024). According to
the Lower Forestry Authority of the State of Brandenburg (2023), a high number of fires were caused by intentional arson and
other anthropogenic actions such as open fires or smoking (see Table S 2 in the Supplement). Therefore, climatic conditions
may not have contributed to the emergence of those fires in a significant way. Furthermore, meteorological projections assume
that air temperatures will increase overall. However, the input data used for this study shows increased precipitation patterns
in Brandenburg in the future scenarios compared to the periods of June 2016 and June 2022 as well (see Fig. S 1 and S 2 in
the Supplement). Consequently, this change in precipitation patterns shown by the input data may have lowered future FFS
in the study region, thus outweighing the effect of higher air temperatures and contributing to the lower mean FFS in future
scenarios compared to the extremely hot and dry year of 2022. The German Weather Service (DWD, 2019) predicts changes
between - 4 % to + 13 % in the annual precipitation sums until the end of the 21st century, illustrating the uncertainty of future
precipitation predictions. As a result, in case of a decrease in precipitation until the end of the 21st century, this will strongly

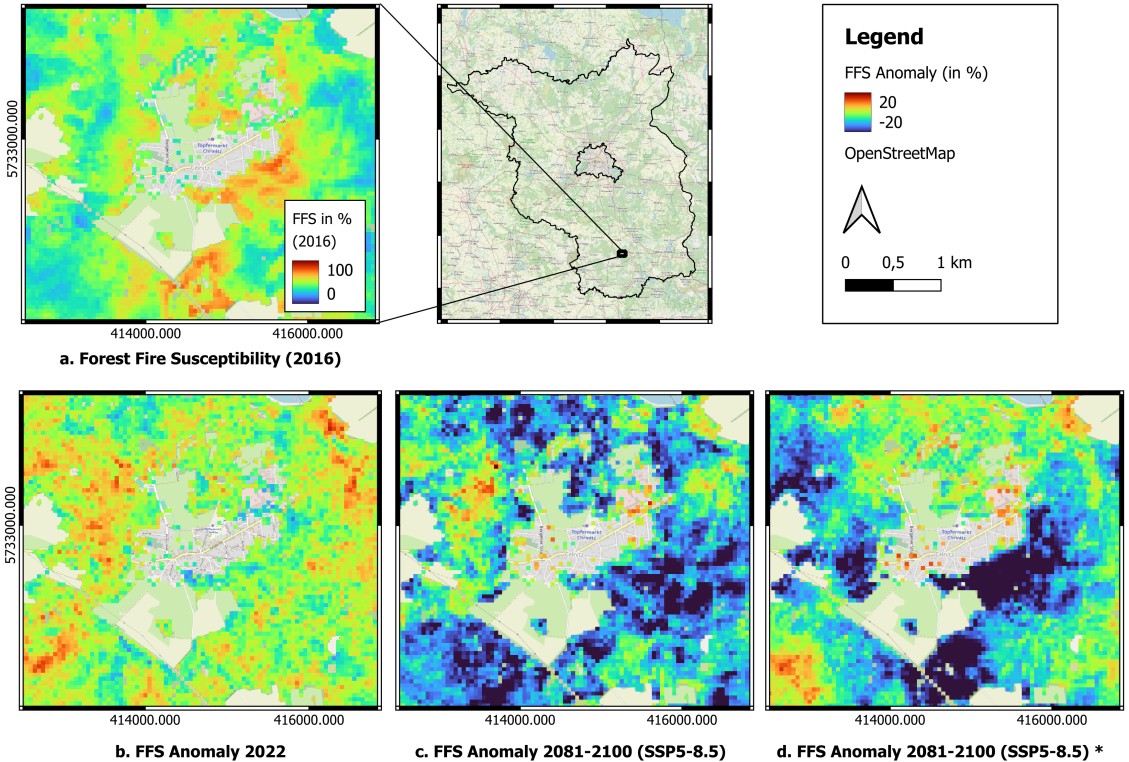

**Figure 7.** Detailed maps of FFS anomalies in the municipality of Crinitz (Brandenburg). Scenarios c and d both show predicted FFS in June 2081-2100 under SSP5-8.5. Scenario d includes projected land cover data, whereas scenario c does not. Base Map © OpenStreetMap contributors 2024. Distributed under the Open Data Commons Open Database License (ODbL) v1.0, Border layer © 2018-2022 GADM.

affect the flammability of Brandenburg's forests and thus the future FFS.

Extreme weather events may be a better indicator of future FFS rather than averaged long-term meteorological trends. Extreme weather conditions such as the dry conditions in 2022 were efficiently captured by the current meteorological data, whereas the multi-annually aggregated monthly projected meteorological data (WorldClim) did not reflect these extreme weather events. For instance, the monthly average precipitation sum in Brandenburg shows flatter curves for the future precipitation, whereas more intense changes in mean precipitation values can be seen in 2016 and 2022 (see Fig. S 2 in the Supplement). For example, the precipitation curve for 2022 shows a substantial drop in March, reflecting a very dry month with low precipitation that may have driven the higher FFS mean value in 2022 compared to other scenarios. Hence, future FFS might turn out higher in reality, given the expected increase in extreme weather events that will enhance the likelihood of drought conditions (Rad et al., 2021; Silva et al., 2018; Wu et al., 2021). To assess the future development of FFS on a local scale, climatic data with a higher temporal resolution is needed to reflect weather extremes more adequately than multi-annually aggregated climate data. The moderate to low influence of topographic predictors in predicting FFS is most likely due to the rather homogeneous to-

pography in Brandenburg. For vegetation parameters, the percentage of broadleaf forest was most important for the modelling. This result aligns with several studies that have shown monocultural coniferous forests to be more sensitive to forest fires (Afreen et al., 2011; Bauhus et al., 2017; Gnilke et al., 2022). Being dominated by pine trees makes Brandenburg particularly susceptible to forest fires. For example, Gnilke et al. (2022) assessed the fire damage in pine forests in Brandenburg, concluding that pure pine stands showed the most burning marks, whereas mixed tree stands were more resilient to forest fires. Furthermore, Buras et al. (2018) have underlined the vulnerability of pine trees located at forest edges, similarly to our results about the influence of distance to forest edge (mean distance for fire points 148.5 m and mean distance for non-fire points 174.8 m; also see Table S 6 in the Supplement). Thus, forest edges in Brandenburg may require special protection to avoid future forest fires.

On a regional scale, anthropogenic parameters appear to be more relevant for FFS. In particular, the distance to urban settlements and railways showed a high significance for modelling FFS in Brandenburg. This confirms the statistics of forest fire emergence in Brandenburg provided by the Lower Forestry Authority of the State of Brandenburg (2023) (see Table S 2 in the Supplement), highlighting that most forest fires in Brandenburg emerge from human negligence or malicious arson. Several other studies have reached the same conclusion (Busico et al., 2019; Cilli et al., 2022; Ghorbanzadeh et al., 2019; Gnilke and Sanders, 2021; He et al., 2022; Ruffault and Mouillot, 2017). However, the distance to military sites only moderately influenced the RF models (see Fig. 3). Furthermore, the Wilcoxon test (see Table S 6 in the Supplement) was not significant, underlining that there was no clear difference in the distribution of fire and non-fire points across Brandenburg. Therefore, the data and model results do not show a clear relationship between distance to military sites and FFS.

## 4.2 Assessing current and future forest fire susceptibility

Overall, the future scenario 2081-2100 (excl. projected land cover data) revealed a substantial increase in mean FFS compared to 2016. However, in 2022 the mean FFS was higher than in 2016 and the two future scenarios. The comparatively high mean FFS of 2022 can be explained by significantly drier and hotter conditions compared to 2016. Nevertheless, the mean FFS value of the future scenario neglecting land cover changes is only slightly below the mean FFS value of 2022 and higher than the mean FFS value of 2016. Considering exclusively future climatic conditions, this indicates an expected overall increase in FFS in Brandenburg until the end of the 21st century compared to June 2016. However, since the future modelled climate data relies on multi-annual monthly averages of air temperature and precipitation, future FFS is possibly underestimated in this study.

The second future scenario including both projected land cover changes (*) and future climatic conditions paints a different picture. As shown in Table 3, mean FFS was lowest of all scenarios indicating an overall decrease in FFS. This result can most likely be explained by two aspects: First, Esri's "Land Cover 2050 - Global" data set (Esri Environment, 2021) used to plot future distance to urban settlements projects a decrease in urbanised areas in the future compared to the Impervious Built-up data set (European Environment Agency [EEA], 2020b). Shrinking urban areas can be explained by demographic changes, such as the ageing and decline of the German population, especially in the East of Germany (Kroll and Haase, 2010). Although Kroll and Haase (2010) state that the ageing of the German population has not yet influenced land use changes, they argue that this is likely to change in the future. Second, Esri's "Land Cover 2050 - Global" data set (Esri Environment, 2021) has

a lower spatial resolution (300 m) than the COPERNICUS Imperviousness data set (European Environment Agency [EEA], 2020b) used to map the distance to "current" urban settlements (10 m). As a result, Esri's data set may show some inaccuracies due to mixed pixel effects. For instance, some smaller settlements may not appear in the future land cover data set. Our results underscore how the inclusion of projected land cover data significantly changes the projected FFS in the future, an aspect that can be further explored in future studies with new land cover projections.

Based on our findings, it can be argued that future urban development trends will significantly influence FFS. Hence, a population decline and abandonment of villages and rural areas may decrease FFS in those areas. However, new settlements due to continuous suburbanization processes may require additional forest fire prevention efforts in the future. Regardless of these trends, the expected increase in drought events in Brandenburg (Gnilke et al., 2022) may intensify the FFS in Brandenburg in the future. Consequently, effective forest fire management strategies in Brandenburg need to address these aspects. Therefore, the following chapter provides key strategies for the management of forest fires in the future.

### 4.3  Strategies for forest fire management in Brandenburg

Forest fire management strategies include the improvement of forest fire prediction, prevention, detection, extinction, constant monitoring of meteorological conditions, and assessment of previous forest fires to improve management strategies (Martell, 2007). An effective forest fire prevention strategy in Brandenburg involves promoting the growth of mixed forests instead of the prevalent monocultural pine forests. In particular, increasing the percentage of broadleaf trees is needed (Ministry for Rural Development, Environment and Agriculture in Brandenburg, 2024; Gnilke et al., 2022). Protection measures should put particular emphasis on forest edges and forests in proximity to any type of anthropogenic infrastructure. The prediction of FFS as implemented here provides a helpful tool to identify the most susceptible forest areas in Brandenburg, where the implementation of forest fire management strategies is most important. Complementing with constant monitoring of meteorological conditions, it can provide a powerful means to predict FFS and to provide an early warning system for forest fires. In addition to that, the constantly updated meteorological data, as well as drought indices and the forest fire danger index provided by the German Meteorological Service (GMS) are essential to predict FFS in Brandenburg (Fekete and Nehren, 2023).

The conventional approach to fire detection involves integrating public reports with observation towers and aerial patrols (Martell, 2007). Increasing the number of observation towers in forest areas with high FFS could speed up fire detection and extinguishment. A valuable forest fire prevention measure is the restriction of human activities in forests or the closure of forests to the public in accordance with the meteorological conditions, given the large anthropogenic contribution to FFS. This is recommendable especially in forest areas with high FFS to decrease the number of fires caused by anthropogenic influences. However, the meaning of forests for recreational purposes, as well as the economic factor of touristic forest users should be considered before implementing such measures. Additionally, implementing public education initiatives on forest fires through school programs and media campaigns is imperative for fostering greater awareness on forest fires and modifying behaviors to reduce ignition risks (Martell, 2007).

Moreover, the implementation of fire breaks is recommendable to limit the spread of forest fires (Berčák et al., 2023). Another strategy can be the thinning of pine forests to reduce fire risk. For example, Crecente-Campo et al. (2009) have concluded

that thinning of *Pinus sylvestris* can contribute to the growth of a mixed leaved forest that has shown to be more resilient to forest fires (Afreen et al., 2011; Bauhus et al., 2017; Gnilke et al., 2022). Finally, it is crucial to employ interregional forest fire management strategies, since forest fires, such as the fire in Bohemian Switzerland National Park in 2022, may spread from neighbouring countries to Germany or vice versa (Boháč and Drápela, 2023). Considering the high FFS in the southeast of

440 the federal state, forest fire management authorities in Brandenburg should consider closer cooperation with the neighbouring country Poland to develop and implement joint management strategies.

## 4.4 Shortcomings and future perspectives

Analysing FFS on a local scale ideally requires climatic data at both high spatial and temporal resolution. High temporal resolution meteorological data better reflects extreme weather events such as droughts. Consequently, the availability of climatic

data at both high spatial and temporal resolution may significantly enhance the quality of future FFS assessments. Ideally, future FFS analysis should incorporate projected climate data with a monthly temporal resolution to reflect future drought events more effectively. Similarly, forest fire products based on remote sensing data with a high spatial and temporal resolution would strongly improve forest fire assessments on smaller scales. However, this type of data is not available yet and its development is limited by the fact that current satellites used for meteorological observations are not able to create images both at high spatial

and temporal resolution due to technical restrictions (Kussul et al., 2023). Forest fire data providers such as the European Forest Fire Information System (EFFIS) supply frequently updated burnt areas for Europe, the Middle East, and North Africa, which is helpful for forest fire analysis on national or international scales. However, the EFFIS burnt areas product is based on the 250 m spatial resolution of MODIS' optical scanner, resulting in smaller forest fires not being included (Achour et al., 2022). Thus, this product is not appropriate for the assessment of FFS at smaller scales.

In a similar vein, an analysis of forest fire detection systems by Barmpoutis et al. (2020) underlines the limitations of satellites in providing both high temporal and spatial resolution. Although satellites such as MODIS or Landsat have thermal infrared bands that can serve for active fire detection, those satellites have their limitations. MODIS has a high temporal resolution, but a spatial resolution of only 1 km for the thermal infrared bands. Landsat satellites, on the other hand, provide higher spatial resolution data (e.g., 100 m for the thermal infrared band for Landsat 8 and 9), but are limited to a temporal resolution of

16 days (Acharya and Yang, 2015; Chanthiya and Kalaivani, 2021; Fu et al., 2020). However, new developments of real time detection and life tracking of wildfires based on a set of over 20 satellites such as provided by OroraTech (OroraTech, 2021) show the potential of future analysis of forest fires.

Nevertheless, it is crucial that local forest fire management institutions provide data on smaller fires as well. However, in the case of the Lower Forestry Authority of the State of Brandenburg, forest fire data was not provided in the form of polygons of

465 burnt areas, but in the form of fire ignition points. Despite the fact that the burnt area (ha) was provided, the exact extent of it could only be assumed. Consequently, model results of FFS prediction might have been more accurate if the actual extent of the forest fires had been available. Nevertheless, with continuous advances in remote sensing, forest fire data may be openly available at higher spatial resolutions in the future, which represents a significant potential for future FFS predictions on a local scale.

Apart from the spatial resolution of forest fire products, the modelling approach to predict FFS should be carefully selected. As previously discussed, meteorological parameters did not have a significant influence on the model. Therefore, future research may consider applying a Long Short-Term Memory (LSTM) model to better incorporate meteorological trends and to improve the understanding of how forests react to droughts and heat waves (Burge et al., 2021; Natekar et al., 2021).

Furthermore, the future land cover change data set (Esri Environment, 2021) had some limitations. First, it only included information on "Artificial Surface or Urban Area". Consequently, a differentiation of different anthropogenic land uses (e.g., campsites, streets, urban settlements, or railways) for the future scenarios was not possible. Instead, the data set was only used to project the future distance to urban settlements. Second, the projection of the data set was only available for 2050. Ideally, a data set reflecting the land use changes until the end of the 21st century would have led to more accurate results. Third, compared to the other land use and land cover data sets used in this study, the spatial resolution of the future land cover change data set (Esri Environment, 2021) was relatively coarse. Therefore, the data set may contain some inaccuracies, thus potentially decreasing the accuracy of the future FFS projections. Nevertheless, to our knowledge, this data set had a relatively high spatial resolution compared to other data sets, which is why it was selected for the study. In the end, the expansion of renewable energies (Hilker et al., 2024), the settlement of new companies and factories (e.g., Tesla gigafactory in Grünheide) (Kühn, 2023), suburbanization processes around Berlin driven by rising housing prices (Leibert et al., 2022), and finally the abandonment of smaller villages due to ageing and population decline is likely to lead to future land cover changes and either heightened or decreased pressures on forests. Consequently, including this data set into the analysis provides valuable information on potential land cover changes. Future research may consider including higher-spatial-resolution land cover change data to model FFS.

Finally, future FFS research may integrate further predictors, dynamic predictors in particular, into their analysis. Following Rad et al. (2021), key variables shaping drought conditions are precipitation, soil moisture, and stream flow. Thus, it may be beneficial to include especially soil moisture data into future analyses. However, due to a lack of soil moisture projections, this parameter was not integrated into this study.

## 5 Conclusions

This study successfully predicted FFS on a regional scale in the federal state of Brandenburg under different scenarios with the RF ML algorithm. The FFS maps show a high FFS in the south and south-east of the federal state. Considering only future meteorological conditions, future FFS is expected to increase compared to the 2016 reference scenario. Extreme events such as droughts can significantly intensify FFS, which was demonstrated by the higher mean FFS value of 2022 compared to the other scenarios. However, including both projected land cover change and future meteorological data into the future projections showed a decrease in FFS. This trend might be driven by demographic changes ultimately leading to future land use changes. The selection of a three-month temporal aggregation of the meteorological data sets was appropriate to reflect long-term meteorological trends. Using climate data at a higher temporal resolution would have shown the effect of extreme weather events more adequately. Therefore, future research could aim at integrating climate data at higher temporal resolution (e.g., weekly)

to integrate the effect of extreme weather events into the predictions.

Our study emphasised the importance of anthropogenic predictors such as distance to urban settlements, railways or campsites. Thus, it is crucial to protect forests from anthropogenic influences to reduce the occurence of forest fires, especially during drought events. Furthermore, we showed the higher resilience of mixed forests in contrast to monocultural forests, confirming previous literature. Forest managers should therefore prioritise the growth of broadleaf trees. Soil parameters such as percentage of silt and sand had a medium to high importance, suggesting that pore sizes and the consequent capacity of the soil to carry and maintain water restricts the availability of water for trees. Finally, topographic parameters such as slope or TWI had a rather low importance for predicting FFS in Brandenburg, which is likely due to the overall rather flat topography of the federal state.

This study and FFS maps can serve local forest managers and firefighters in the prevention of forest fires in the region. Furthermore, the identification of contributing variables can support the development of forest fire management strategies adapted to local circumstances.

*Code availability.* The code used for this study, as well as the forest fire susceptibility maps (Figs. 4 to 7) are publicly available on Zenodo at https://doi.org/10.5281/zenodo.14214917 and on GitHub at https://github.com/ka-horn/forest-fire-susceptibility-modelling.

*Data availability.* The forest fire data was provided by the Lower Forestry Authority of the State of Brandenburg and can be acquired upon request at the Eberswalde Forestry Competence Center (EFCC).

*Author contributions.* **Katharina H. Horn**: Writing – original draft, Visualization, Validation, Software, Methodology, Investigation, Formal analysis, Data curation. **Stenka Vulova**: Writing – review & editing, Supervision, Methodology, Conceptualization. **Hanyu Li**: Writing – review & editing, Methodology, Conceptualization. **Birgit Kleinschmit**: Writing – review & editing, Supervision, Resources, Project administration, Funding acquisition, Conceptualization.

*Competing interests.* The authors declare that they have no known competing financial interests or personal relationships that could have appeared to influence the work reported in this paper.

*Acknowledgements.* This investigation was funded through the Einstein Research Unit 'Climate and Water under Change' from the Einstein Foundation Berlin and Berlin University Alliance (ERU-2020-609). We would also like to thank Chunyan Xu, Christian Schulz, and Pedro Alencar for their support on technical and conceptual matters. For the scope of this research, Chat-GPT 3.5 was used to support coding, writing, and editing of this article. We acknowledge support by the Open Access Publication Fund of TU Berlin.

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
