# Peer review of "Modelling Current and Future Forest Fire Susceptibility in north-east Germany"

_EGUsphere, 2024_

## Author Comment (AC1)

**Response to anonymous reviewer comment RC1**

| | |
|---|---|
| **Authors** | Katharina Horn, Stenka Vulova, Hanyu Li, Birgit Kleinschmit |
| **Title** | Modelling Forest Fire Susceptibility in Brandenburg under Current and Future Scenarios |
| **Manuscript type** | Special Issue: Current and future water-related risks in the Berlin–Brandenburg region |
| **Journal** | Natural Hazards and Earth System Sciences (NHESS) |
| **Handling editor** | Axel Bronstert, axelbron@uni-potsdam.de |
| **Citation of reviewer comment** | https://doi.org/10.5194/egusphere-2024-1380-RC1 |

*Reviewer comments are marked with "RC1" and answers from the authors are marked with "A" in cursive. When applicable, we provided quotations of the modified sections from the manuscript to further illustrate the response. The modified sections that are quoted here are marked in **bold**.*

*We would like to warmly thank the reviewer for the extensive and qualitative feedback. We hope that the integration of the reviewer's feedback has improved the quality of the manuscript.*
* * *
**Reviewer Comment #1**

RC1: Thank you very much for submitting this very interesting manuscript. The authors modelled current and future forest fire susceptibility in Brandenburg using a random forest approach. They analyzed variable importances of topographic, climatic, anthropogenic, soil and vegetation predictors, highlighting the influence of human factors for fire ignitions in Brandenburg. Overall, one strength of this article is the comprehensive description of methods and its well written nature. I very much enjoyed reading the study. Well done! So far, I only have only one major comment regarding the temporal selection of climatic variables, the rest is minor.

*A: Thank you very much for the positive feedback on our manuscript and your careful examination. In the following, we will provide some explanatory comments in response to your comments.*

RC1: Major comment: Over that whole manuscript I am wondering why only a selection of months (here June) was used to build your random forest models. I agree that human factors have a strong influence on fire susceptibility in Brandenburg and I can also follow your discussion on explaining the rather weak influence of climate variables given your analysis and

missing extreme events in the data. However, in general, I miss more details/justification why the selection of only few summer months was done here. In the variable description the authors refer to a publication by He et al. (2022), which however, modelled Australian bushfires, thus the climate-fire-susceptibility relations might be different from those compared to forest fires in Brandenburg. Therefore, I kindly ask the authors at least to better justify the rather strong assumption to select only certain months for their analysis. I also kindly ask the authors to test if your random forest analysis yields very different results if you include all the months of the year (from Table 1 I assume that monthly resolution is given also for the future data). After all, I think including more months in your analysis is highly valuable, because this could also improve our predictive outcomes and messages you could convey for your future projections (as you discussed in section 4.2). I suspect the main reason why your future predictions are weakly diverging from the present day, might not only be due to a limited representation of extreme events in our future data, but rather the fact that the only changing variables in your predictions are climatic - and those have a fairly weak importance our RF-models.

*A: We thank the reviewer for this detailed feedback. For Random Forest model training, we included forest fire data from all available months of all years for the analysis (2014 to 2022). Respectively, we included climatic data of all available months of the mentioned time period for the model training. The month of June was selected for model prediction exclusively. We decided upon this month after carefully assessing the forest fire data provided by the Lower Forestry Authority of the State of Brandenburg (2023), which showed that the majority of forest fire events occurred in the month of June based on the years of 2014 to 2022.*

*To clarify this matter, we modified the following section at the end of chapter 2.2 (ll. 92 f.):*

*"After analysing the monthly frequency of forest fires in the federal state of Brandenburg, the month of June was selected for the prediction of the four scenarios, since forest fire data showed the highest number of forest fires in this month between 2014 to 2022 (Lower Forestry Authority of the State of Brandenburg, 2023).* **For model training, we used all available forest fire events of all months between 2014 to 2022 and pre-processed monthly climatic data sets in accordance with the available forest fire data."**

*Regarding your statement* "I suspect the main reason why your future predictions are weakly diverging from the present day, might not only be due to a limited representation of extreme events in our future data, but rather the fact that the only changing variables in your predictions are climatic - and those have a fairly weak importance our RF-models.": *Thank you very much for pointing this out. We absolutely agree with this interpretation. Since the second reviewer expressed a similar criticism, we did some more research into available data and discovered the data set "Land Cover 2050 - Global" by Esri Environment (2021), which predicted global land cover change for the year of 2050. It was used to compute future proximity to urban settlements. The layer of future proximity to urban settlements was then used for the future prediction of forest fire susceptibility. For further explanations on this aspect, please refer to the answers to the second review comment RC2.*

*Reference:*

*Esri Environment. (2021). Land Cover 2050—Global.*
*https://hub.arcgis.com/datasets/esri::land-cover-2050-global/about*

RC1: Minor comments: Line 5: Please shortly define fire susceptibility already in the abstract.

*A: Thank you for this suggestion. We modified the abstract accordingly. This is the updated abstract:*

**"Preventing and fighting forest fires has been a challenge worldwide in recent decades. Forest fires alter forest structure and composition, threaten people's livelihoods, and lead to economic losses, as well as soil erosion and desertification. Climate change and related drought events, paired with anthropogenic activities, have magnified the intensity and frequency of forest fires. Consequently, we analysed forest fire susceptibility (FFS), which can be understood as the likelihood of fire occurrence in a certain area. We applied Random Forest (RF) machine learning (ML) algorithm to model current and future FFS in the federal state of Brandenburg (Germany) using topographic, climatic, anthropogenic, soil, and vegetation predictors. FFS was modelled at a spatial resolution of 50 metres for current (2014-2022) and future scenarios (2081-2100). Model accuracy ranged between 69 % ($RF_{test}$) and 71 % (LOYO), showing a moderately high model reliability for predicting FFS. The model results underscore the importance of anthropogenic parameters and vegetation parameters in modelling FFS on a regional level. This study will allow forest managers and environmental planners to identify areas, which are most susceptible to forest fires, enhancing warning systems and prevention measures."**

RC1: Line 16: Consider to check the reference for the increasing number of fires in Germany. I guess it should be rather the study by Gnilke et al. from 2021 not 2022.

*A: Thank you for pointing this out. We checked this again and you are right. The updated sentence is now as follows (ll. 14 f.):*

"In Germany, very low precipitation has been occurring more frequently in the last six years, leading to an increased number of forest fires (**Gnilke and Sanders, 2021**)."

RC1: Line 54: Consider to check the reference Gnilke & Sanders 2021. I think here it should be rather the Gnilke et al. 2022 publication.

*A: Thank you for pointing this out. We checked this again and you are right. The updated sentences are now as follows (ll. 51 ff.):*

*"Due to a high percentage of coniferous forest, this federal state has been particularly prone to forest fires in the past. Furthermore, remnants of old munitions at former military training sites have been causing forest fires in Brandenburg in 2018 and 2019 (**Gnilke et al., 2022**)."*

RC1: Line 62: Could you please cite some of the few studies that you found, which have analyzed current and future FFS at a high spatial resolution?

*A: Thank you for this useful suggestion. The studies mapping current forest fire susceptibility at smaller scales and relatively high spatial resolution are Ghorbanzadeh et al. (2019), Pourtaghi et al. (2014), Razavi-Termeh et al. (2020), and Suryabhagavan et al. (2016).*

*We checked this text passage and our references again but could not find any studies that modeled future forest fire susceptibility at a small scale and high spatial resolution. Accordingly we corrected the sentence in the manuscript as follows (ll. 60 ff.):*

*"To our knowledge, only few studies have analysed FFS at a high spatial resolution so far (**Ghorbanzadeh et al., 2019; Suryabhagavan et al., 2016; Razavi-Termeh et al., 2020; Pourtaghi et al., 2015**) and we do not know of any studies that modelled future FFS at a high spatial resolution."*

*References:*

*Ghorbanzadeh, O., Blaschke, T., Gholamnia, K., & Aryal, J. (2019). Forest Fire Susceptibility and Risk Mapping Using Social/Infrastructural Vulnerability and Environmental Variables. Fire, 2(3), Article 3. https://doi.org/10.3390/fire2030050*

*Pourtaghi, Z. S., Pourghasemi, H. R., & Rossi, M. (2015). Forest fire susceptibility mapping in the Minudasht forests, Golestan province, Iran. Environmental Earth Sciences, 73(4), 1515–1533. https://doi.org/10.1007/s12665-014-3502-4*

*Razavi-Termeh, S. V., Sadeghi-Niaraki, A., & Choi, S.-M. (2020). Ubiquitous GIS-Based Forest Fire Susceptibility Mapping Using Artificial Intelligence Methods. Remote Sensing, 12(10), Article 10. https://doi.org/10.3390/rs12101689*

*Suryabhagavan, K. V., Alemu, M., & Balakrishnan, M. (2016). GIS-based multi-criteria decision analysis for forest fire susceptibility mapping: A case study in Harenna forest, southwestern Ethiopia. Tropical Ecology, 57(1), 33–43.*

RC1: Line 79: I could not find A2 in the supplement. Maybe it should be S2 here.

*A: This is absolutely true. We corrected this information accordingly. The updated sentence is now as follows (ll. 78 ff.):*

*"To represent the current state, the years of 2016 and 2022 were selected after carefully analysing the monthly precipitation sums and mean monthly air temperature of Brandenburg between 2014 to 2022 (see Fig. S 1 and **S 2** in the Supplement)."*

RC1: Line 115: Here the authors state that climatic variables where aggregated to 3 months, but in line 90 is written that only June was selected. Please indicate which months were used to train the models. (see also my major concern) If only June was selected to built the RFs, I recommend to check if the peak fire season might be shifted under future climate conditions - and if so, shortly discuss this point in the discussion.

*A: Thank you for your comment. Please refer to our answer provided to the "Major Comment", where we provided an answer on this matter.*

RC1: Line 320: I agree to the points you raised to explain the weak importance of climatic variables. However, I miss a discussion what would happen if more months (and therefore more intra-annual variability) were considered in your approach (see major concern). How would that change your results?

*A: Considering the fact that we used data from all months of 2014 to 2022 for model training, we believe that it is not necessary to reflect on this here any further.*

RC1: Line 369: I highly acknowledge that you outline forest fire prevention strategies in Brandenburg. Please add references for the lines 369 – 371.

*A: Thank you for pointing out the lack of references here. We added those accordingly. This is the updated text passage (ll. 398 ff.):*

*"An effective forest fire prevention strategy in Brandenburg involves promoting the growth of mixed forests instead of the prevalent monocultural pine forests. In particular, increasing the percentage of broadleaf trees is needed (**Ministry for Rural Development, Environment and Agriculture in Brandenburg, 2024; Gnilke et al., 2022**)."*

*References:*

*Gnilke, A., Liesegang, J., & Sanders, T. (2022). Potential forest fire prevention by management-An analysis of fire damage in pine forests. https://literatur.thuenen.de/digbib_extern/dn065237.pdf*

*Ministry for Rural Development, Environment and Agriculture in Brandenburg. (2024). Strategie des Landes Brandenburg zur Anpassung an die Folgen des Klimawandels. https://mluk.brandenburg.de/sixcms/media.php/9/Klimaanpassungsstrategie-BB-Kurzfassung.pdf*

---

## Referee Report (RR1)

Thank you very much for resubmitting this very interesting manuscript and fully addressing my concerns. I think that including changes in land cover has greatly improved the manuscript. Well done! I have only one minor, rather technical comment. Other than that, I recommend this article for publication.

Minor comment:

- Lines: 348 – 351:

There it is written:
*Furthermore, meteorological projections assume that air temperatures will increase overall. However, precipitation is expected to increase in the future as well (see Fig. S 1 and S 2 in the Supplement). Consequently, wetter conditions may have lowered future FFS, outweighing the effect of higher air temperatures and contributing to the lower mean FFS in future scenarios compared to the extremely hot and dry year of 2022.*

I think here the wording regarding the wetter conditions is "too strong". In Fig. S1 and S2 you are technically comparing different data sources (DWD data and GCM data). So I think that concluding only from Fig. S1 and S2 that future precipitation is expected to increase is too strong. For instance, it could be also that MPI-ESM-1-2-HR is generally overestimating precipitation in June in this region. So the increase in precipitation could be mostly an effect of a general bias in the GCM compared to the station data from DWD. Although it becomes clearer that there are larger uncertainties in future precipitation patterns later, I would avoid this general statement here. Maybe it could help to rewrite those sentences clearly saying that this (only) refers to the meteorological input data that you are using.

---

## Author Response (AR2)

**Response to anonymous reviewer comment RC1 (October 9th, 2024)**

| Authors | Katharina Horn, Stenka Vulova, Hanyu Li, Birgit Kleinschmit |
|---|---|
| **Title** | Modelling Forest Fire Susceptibility in Brandenburg under Current and Future Scenarios |
| **Manuscript type** | Special Issue: Current and future water-related risks in the Berlin–Brandenburg region |
| **Journal** | Natural Hazards and Earth System Sciences (NHESS) |
| **Handling editor** | Axel Bronstert, axelbron@uni-potsdam.de |
| **Citation of reviewer comment** | https://doi.org/10.5194/egusphere-2024-1380-RC1 |

*Reviewer comments are marked with "RC1" and answers from the authors are marked with "A" in cursive. When applicable, we provided quotations of the modified sections from the manuscript to further illustrate the response. The modified sections that are quoted here are marked in **bold**.*

*We would like to warmly thank the reviewer for the extensive and qualitative feedback. We hope that the integration of the reviewer's feedback has improved the quality of the manuscript.*
* * *
**Reviewer Comment #1**

RC1: Thank you very much for submitting this very interesting manuscript. The authors modelled current and future forest fire susceptibility in Brandenburg using a random forest approach. They analyzed variable importances of topographic, climatic, anthropogenic, soil and vegetation predictors, highlighting the influence of human factors for fire ignitions in Brandenburg. Overall, one strength of this article is the comprehensive description of methods and its well written nature. I very much enjoyed reading the study. Well done! So far, I only have only one major comment regarding the temporal selection of climatic variables, the rest is minor.

*A: Thank you very much for the positive feedback on our manuscript and your careful examination. In the following, we will provide some explanatory comments in response to your comments.*

RC1: Major comment: Over that whole manuscript I am wondering why only a selection of months (here June) was used to build your random forest models. I agree that human factors have a strong influence on fire susceptibility in Brandenburg and I can also follow your discussion on explaining the rather weak influence of climate variables given your analysis and

missing extreme events in the data. However, in general, I miss more details/justification why the selection of only few summer months was done here. In the variable description the authors refer to a publication by He et al. (2022), which however, modelled Australian bushfires, thus the climate-fire-susceptibility relations might be different from those compared to forest fires in Brandenburg. Therefore, I kindly ask the authors at least to better justify the rather strong assumption to select only certain months for their analysis. I also kindly ask the authors to test if your random forest analysis yields very different results if you include all the months of the year (from Table 1 I assume that monthly resolution is given also for the future data). After all, I think including more months in your analysis is highly valuable, because this could also improve our predictive outcomes and messages you could convey for your future projections (as you discussed in section 4.2). I suspect the main reason why your future predictions are weakly diverging from the present day, might not only be due to a limited representation of extreme events in our future data, but rather the fact that the only changing variables in your predictions are climatic - and those have a fairly weak importance our RF-models.

*A: We thank the reviewer for this detailed feedback. For Random Forest model training, we included forest fire data from all available months of all years for the analysis (2014 to 2022). Respectively, we included climatic data of all available months of the mentioned time period for the model training. The month of June was selected for model prediction exclusively. We decided upon this month after carefully assessing the forest fire data provided by the Lower Forestry Authority of the State of Brandenburg (2023), which showed that the majority of forest fire events occurred in the month of June based on the years of 2014 to 2022.*

*To clarify this matter, we modified the following section at the end of chapter 2.2 (ll. 97 f.):*

*"After analysing the monthly frequency of forest fires in the federal state of Brandenburg, the month of June was selected for the prediction of the four scenarios, since forest fire data showed the highest number of forest fires in this month between 2014 to 2022 (Lower Forestry Authority of the State of Brandenburg, 2023). **For model training, we used all available forest fire events of all months between 2014 to 2022 and pre-processed monthly climatic data sets in accordance with the available forest fire data.**"*

*Regarding your statement* "I suspect the main reason why your future predictions are weakly diverging from the present day, might not only be due to a limited representation of extreme events in our future data, but rather the fact that the only changing variables in your predictions are climatic - and those have a fairly weak importance our RF-models.": *Thank you very much for pointing this out. We absolutely agree with this interpretation. Since the second reviewer expressed a similar criticism, we did some more research into available data and discovered the data set "Land Cover 2050 - Global" by Esri Environment (2021), which predicted global land cover change for the year of 2050. It was used to compute future proximity to urban settlements. The layer of future proximity to urban settlements was then used for the future prediction of forest fire susceptibility. For further explanations on this aspect, please refer to the answers to the second review comment RC2.*

*Reference:*

*Esri Environment. (2021). Land Cover 2050—Global.*
*https://hub.arcgis.com/datasets/esri::land-cover-2050-global/about*

RC1: Minor comments: Line 5: Please shortly define fire susceptibility already in the abstract.

*A: Thank you for this suggestion. We modified the abstract accordingly. This is the updated abstract:*

*"**Preventing and fighting forest fires has been a challenge worldwide in recent decades. Forest fires alter forest structure and composition, threaten people's livelihoods, and lead to economic losses, as well as soil erosion and desertification. Climate change and related drought events, paired with anthropogenic activities, have magnified the intensity and frequency of forest fires. Consequently, we analysed forest fire susceptibility (FFS), which can be understood as the likelihood of fire occurrence in a certain area. We applied Random Forest (RF) machine learning (ML) algorithm to model current and future FFS in the federal state of Brandenburg (Germany) using topographic, climatic, anthropogenic, soil, and vegetation predictors. FFS was modelled at a spatial resolution of 50 metres for current (2014-2022) and future scenarios (2081-2100). Model accuracy ranged between 69 % (RF_{test}) and 71 % (LOYO), showing a moderately high model reliability for predicting FFS. The model results underscore the importance of anthropogenic parameters and vegetation parameters in modelling FFS on a regional level. This study will allow forest managers and environmental planners to identify areas, which are most susceptible to forest fires, enhancing warning systems and prevention measures.**"*

RC1: Line 16: Consider to check the reference for the increasing number of fires in Germany. I guess it should be rather the study by Gnilke et al. from 2021 not 2022.

*A: Thank you for pointing this out. We checked this again and you are right. The updated sentence is now as follows (ll. 14 f.):*

*"In Germany, very low precipitation has been occurring more frequently in the last six years, leading to an increased number of forest fires (**Gnilke and Sanders, 2021**)."*

RC1: Line 54: Consider to check the reference Gnilke & Sanders 2021. I think here it should be rather the Gnilke et al. 2022 publication.

*A: Thank you for pointing this out. We checked this again and you are right. The updated sentences are now as follows (ll. 51 ff.):*

*"Due to a high percentage of coniferous forest, this federal state has been particularly prone to forest fires in the past. Furthermore, remnants of old munitions at former military training sites have been causing forest fires in Brandenburg in 2018 and 2019 (**Gnilke et al., 2022**)."*

RC1: Line 62: Could you please cite some of the few studies that you found, which have analyzed current and future FFS at a high spatial resolution?

*A: Thank you for this useful suggestion. The studies mapping current forest fire susceptibility at smaller scales and relatively high spatial resolution are Ghorbanzadeh et al. (2019), Pourtaghi et al. (2014), Razavi-Termeh et al. (2020), and Suryabhagavan et al. (2016).*

*We checked this text passage and our references again but could not find any studies that modeled future forest fire susceptibility at a small scale and high spatial resolution. Accordingly we corrected the sentence in the manuscript as follows (ll. 60 ff.):*

*"To our knowledge, only few studies have analysed FFS at a high spatial resolution so far (**Ghorbanzadeh et al., 2019; Suryabhagavan et al., 2016; Razavi-Termeh et al., 2020; Pourtaghi et al., 2015) and we do not know of any studies that modelled future FFS at a high spatial resolution.**"*


| Authors | Katharina Horn, Stenka Vulova, Hanyu Li, Birgit Kleinschmit |
|---|---|
| **Title** | Modelling Forest Fire Susceptibility in Brandenburg under Current and Future Scenarios |
| **Manuscript type** | Special Issue: Current and future water-related risks in the Berlin–Brandenburg region |
| **Journal** | Natural Hazards and Earth System Sciences (NHESS) |
| **Handling editor** | Axel Bronstert, axelbron@uni-potsdam.de |
| **Citation of reviewer comment** | https://doi.org/10.5194/egusphere-2024-1380-RC2 |

*Reviewer comments are marked with "RC1" and answers from the authors are marked with "A" in cursive. When applicable, we provided quotations of the modified sections from the manuscript to further illustrate the response. The modified sections that are quoted here are marked in **bold**.*

*We would like to warmly thank the reviewer for the extensive and qualitative feedback. We hope that the integration of the reviewer's feedback has improved the quality of the manuscript.*
* * *
**Reviewer Comment #2**

RC2: The manuscript "Modelling Current and Future Forest Fire Susceptibility in north-east Germany" by Horn et al. presents an interesting approach by utilizing a variety of predictor variables to model forest fire susceptibility in Brandenburg. The study is well-written and employs state-of-the-art methods and datasets. However, significant methodological flaws heavily influence the results, raising concerns about the validity of the study's conclusions.

*A: We thank the author for the valuable feedback and the appreciation of our work. We respect the criticisms regarding the methodology and the datasets. In the following, we will further elaborate on how we addressed these criticisms to improve the manuscript.*

RC2: Firstly, the study aggregates meteorological variables at a monthly level and further combines them into three-month periods. This coarse temporal resolution is problematic, particularly when attempting to account for the effects of prolonged droughts. The current approach diminishes the impact of very dry periods that end or begin with heavy rainfall. A more appropriate method would be to use a daily or weekly measure that accumulates over time, such as a drought index or a fire weather index. The authors' claim that higher-resolution data is unavailable is outdated, as daily datasets with reasonable resolution,

including wind, humidity, and other relevant variables, are available from sources like the Copernicus Climate Change Service.

*A: We thank the reviewer for the valuable feedback and for pointing out additional data sources. It is true that a finer temporal resolution (e.g. daily, weekly) can better account for certain extreme weather events such as extreme rainfall events or heat waves. However, in the following text, we would like to provide further explanation as to why we decided on a three-month aggregation of the meteorological data.*

*First of all, in order to model and compare forest fire susceptibility under current and future scenarios, the same temporal and spatial resolution of the meteorological data was required to ensure comparability of the results. For our future selected scenarios (2081-2100 under SSP 3.70 and SSP 5.85), meteorological data was only available at a multi-annual monthly scale. Using different temporal resolutions for current and future scenarios (e.g. daily/weekly for the current scenarios and monthly for the future scenarios) would have decreased the comparability of the results. Consequently, in order to align with this temporal resolution, we processed monthly air temperature and precipitation data for both current and future scenarios. The Copernicus Climate Change Service mentioned by the reviewer does provide some very valuable and interesting data sets, such as "Climate extreme indices and heat stress indicators derived from CMIP6 global climate projections" or "Temperature statistics for Europe derived from climate projections". However, the spatial resolution of the data sets is too coarse to account for spatial variances on a regional scale such as in Brandenburg.*

*Second of all, for the scope of this research, we wanted to consider the effect of meteorological droughts on forest fire susceptibility in Brandenburg. To capture this, we decided on a three-month aggregation of the meteorological data. Several authors have used a three-month aggregation of SPEI to identify meteorological droughts: Zhou et al. (2023) have pointed out that a 3-month time lag is best to identify meteorological drought events. In their study, they compute the SPEI, which is a commonly used index to monitor meteorological droughts. Similarly, Petrovic et al. (2022) used a 3-month SPEI to examine droughts in Germany. Other authors have done the same for different regions across the globe (Wen et al. 2020, Guo et al. 2018).*

*Furthermore, a monthly aggregation of meteorological data has been applied by different authors working on forest fires to understand meteorological trends (Busico et al., 2019; He et al. 2022; Wang et al. 2021). He et al. (2022) further suggested using monthly or quarterly meteorological data to investigate fire emergence, which was another reason for the aggregation of air temperature and precipitation data to three months. Therefore, we consider a three-month aggregation of the meteorological data an adequate approach to investigate the relation between air temperature and precipitation patterns and forest fire susceptibility in Brandenburg. In particular, capturing long-term trends prior to the emergence of a forest fire was key to this investigation, which was ensured by using a three-month aggregation of the climatic data.*

*In order to address this decision more clearly in the manuscript, we modified the following paragraph in section 2.3.2 a) Meteorology (ll. 124 ff.):*

*"Following the suggestions by He et al. (2022), we used monthly climate data between 2013 and 2022, which was aggregated to three months to incorporate precipitation and air temperature prior to the occurrence of a forest fire. **Several forest fire related studies have used a monthly aggregation of meteorological data sets to model forest fires (Busico et al., 2019; Wang et al., 2021; He et al., 2022). He et al. (2022) further argue that future studies should consider a monthly or quarterly aggregation of meteorological data when investigating forest fires. Especially in order to identify conditions of meteorological droughts prior to the emergence of a forest fire, we followed the methodology of other authors that used a three-month aggregation of the broadly used SPEI drought index to identify meteorological droughts (Zhou et al., 2023; Wen et al., 2020; Guo et al., 2018).**"*

*"Figure S 6 illustrates the False Positive Rate (FPR) and False Negative Rate (FNR) of the RF models. The grey and black lines show the FPR and FNR, respectively. The red line illustrates the mean air temperature per year, computed based on monthly mean air temperatures. The blue line shows mean precipitation sums per year, based on monthly precipitation sums.*

*The figure shows that across the years, FPR and FNR only differ slightly. Whereas air temperature and precipitation values change more substantially across all years, this did not appear to significantly influence the FPR and FNR. These results underline that the model was not so sensitive to changing weather conditions. Therefore, it can be assumed that the*

*model has more of a spatial than temporal influence. This means that the model is better at distinguishing where forest fires may occur but has more difficulties to understand forest fire prone weather conditions. This drawback could be addressed by future research that could for example utilise a Long Short-Term Memory (LSTM) model. By introducing an internal memory unit known as the 'cell state' and three gate units: the forget gate, input gate, and output gate, LSTM is able to process short and long-term meteorological trends and the subsequent effects on forest fire susceptibility more adequately."*

*To address this in the manuscript, we added the following paragraph (ll. 469 ff.):*

*"Apart from the spatial resolution of forest fire products, the modelling approach to predict FFS should be carefully selected. As previously discussed, meteorological parameters did not have a significant influence on the model. Therefore, future research may consider applying a Long Short-Term Memory (LSTM) model to better incorporate meteorological trends and to improve the understanding of how forests react to droughts and heat waves (Burge et al., 2021; Natekar et al., 2021)."*

*Furthermore, we added the reference to the figure in the Discussion section (ll. 342 f.):*

*"The results reflect that climatic parameters do not appear to play a pivotal role for FFS (see Fig. S 6 in the Supplement)."*

*Additional references:*

*Burge, J., Bonanni, M., Ihme, M., and Hu, L.: Convolutional LSTM Neural Networks for Modeling Wildland Fire Dynamics, https://doi.org/10.48550/arXiv.2012.06679, arXiv:2012.06679 [cs], 2021*

*Natekar, S., Patil, S., Nair, A., and Roychowdhury, S.: Forest Fire Prediction using LSTM, in: 2021 2nd International Conference for Emerging Technology (INCET), pp. 1–5, https://doi.org/10.1109/INCET51464.2021.9456113, 202*
* * *
RC2: Finally, the study's future projections are problematic. The authors adjust only climate variables—despite finding them to have minimal effect—while leaving all other factors static. If the distance to urban settlements is the main driver of fire susceptibility, as the study suggests, then future projections should incorporate urban development trends, not just climate.

*A: We thank the reviewer for pointing this out. We agree with the estimation that the future projections would highly benefit from the inclusion of further dynamic parameters such as future urban development. To address this matter, we researched available data sets of future*

*urban cover. We decided to use Esri's "Land Cover 2050 - Global" data set (Esri Environment, 2021) that models different land cover classes at 300 m spatial resolution on a global scale for the year of 2050 as it had a higher spatial resolution than other data products. We extracted the information for the land cover class "Artificial Surface or Urban Area", resampled it to 50 m spatial resolution and computed the distance to "Artificial surface or Urban Area". Consequently, we decided to replace the scenario 2081-2100 (SSP 3.70) by the scenario June 2081-2100 (SSP 5.85) including this projected land cover change data. Therefore, the two future scenarios presented in the manuscript main text now both refer to the time period of June 2081-2100 under SSP 5.85. The scenario of June 2081-2100 (SSP 3.70) was moved to the Supplement (see Fig. S 10 to S 13).*

*Accordingly, we now present the following four scenarios within the main text of the manuscript:*

- *June 2016,*
- *June 2022,*
- *June 2081-2100 (SSP 5.85), and*
- *June 2081-2100 (SSP 5.85) incl. projected land cover data.*

*Despite the fact that the future land cover data set for 2050 does not cover the same time period as the future climatic data sets, we consider it valuable in reflecting future trends in urban development affecting future FFS. Furthermore, projected land cover data for 2081-2100 was not available; thus, we selected the Esri product due to it being a dataset with an acceptable spatial resolution which was closest to this time period.*

*Since we replaced the future scenario of June 2081-2100 (SSP 3.70) with the new future scenario June 2081-2100 including projected land cover data for 2050, we changed several sections, figures, and tables in the main text and the Supplement.*

*Specifically, we applied the following changes to figures & tables:*

*Table 1: We added the information for the projected land cover data set in the last row of Table 1.*

*Figure 2: We updated and replaced Figure 2.*

[Figure]

*Figure 2. Methodological approach for modelling forest fire susceptibility under different scenarios.*

*Figure 4:* We replaced Figure 4.

[Figure]

**Figure 4. Forest fire susceptibility in Brandenburg under different scenarios. Scenarios c and d both show predicted FFS in June 2081-2100 under SSP 5.85. Scenario d includes projected land cover data, whereas scenario c does not. Border layer © 2018-2022 GADM.**

*Figure 5:* We replaced Figure 5.

[Figure]

*Figure 5. Forest fire anomalies compared to 2016. Scenarios b and c both show predicted FFS in June 2081-2100 under SSP 5.85. Scenario d includes projected land cover data, whereas scenario c does not. Border layer © 2018-2022 GADM.*

*Figure 6: We replaced Figure 6 and moved the previous version to the Supplement (Fig. S 12).*

[Figure]

***Figure 6. Detailed maps of FFS anomalies in the municipality of Medewitz (Brandenburg). Scenarios c and d both show predicted FFS in June 2081-2100 under SSP 5.85. Scenario d includes projected land cover data, whereas scenario c does not. Base Map © OpenStreetMap contributors 2024. Distributed under the Open Data Commons Open Database License (ODbL) v1.0, Border layer © 2018-2022 GADM.***

*Figure 7: We replaced Figure 7 and moved the previous version to the Supplement (Fig. S 13).*

[Figure]

*Figure 7. Detailed maps of FFS anomalies in the municipality of Crinitz (Brandenburg). Scenarios c and d both show predicted FFS in June 2081-2100 under SSP 5.85. Scenario d includes projected land cover data, whereas scenario c does not. Base Map © OpenStreetMap contributors 2024. Distributed under the Open Data Commons Open Database License (ODbL) v1.0, Border layer © 2018-2022 GADM.*

*We updated the results of this table according to the new future predictions including projected land cover data:*

*Table 3. Statistical overview of the four forest fire susceptibility scenarios.* **Scenarios "2081-2100" and "2081-2100 \*" both show predicted FFS in June 2081-2100 under SSP 5.85. Scenario "2081-2100 \*" includes projected land cover data, whereas scenario "2081-2100"** *does not.*

|  | *2016* | *2022* | *2081-2100* | *2081-2100 \** |
|---|---|---|---|---|
| *Minimum* | *0.040* | *0.040* | *0.042* | *0.072* |
| *Maximum* | *0.936* | *0.964* | *0.976* | *0.878* |
| *Mean* | *0.409* | *0.419* | *0.414* | *0.393* |

| | | | | |
|---|---|---|---|---|
| Standard Deviation | 0.147 | 0.146 | **0.144** | **0.116** |

*We added the following figures to the Supplement:*

[Figure]

**Figure S 10. Predicted forest fire susceptibility in June 2081-2100 (SSP 3.70) excluding projected land cover data. Border layer © 2018-2022 GADM.**

[Figure]

*Figure S 11. Forest fire anomaly in June 2081-2100 (SSP 3.70) excluding projected land cover data compared to the reference scenario of June 2016. Border layer © 2018-2022 GADM.*

*We moved the previous versions of Figures 6 and 7 in the main text to the Supplement:*

[Figure]

*Figure S 12. Detailed maps of FFS anomalies in the municipality of Medewitz (Brandenburg). Base Map © OpenStreetMap contributors 2024. The future scenarios were computed excluding projected land cover data. Distributed under the Open Data Commons Open Database License (ODbL) v1.0, Border layer © 2018-2022 GADM.*

[Figure]

**Figure S 13. Detailed maps of FFS anomalies in the municipality of Crinitz (Brandenburg). The future scenarios were computed excluding projected land cover data. Base Map © OpenStreetMap contributors 2024. Distributed under the Open Data Commons Open Database License (ODbL) v1.0, Border layer © 2018-2022 GADM.**

*According to the updated figures and tables, we modified / added the following sections to the main text of the manuscript:*

*Abstract*

*We deleted "considering different shared socioeconomic pathways (SSP 3.70 and SSP 5.85)". Further minor changes to the abstract were made based on the feedback from reviewer #1 (see answer to the first review comment).*

*2.2 Current and future forest fire susceptibility scenarios* *(ll. 85 ff.)*

[revised manuscript text omitted]

RC2: Given these concerns, I must recommend the rejection of this manuscript for publication in its current form. I strongly encourage the authors to revisit and reevaluate their methodology and resubmit the paper once the results and conclusions are more reliable, as the study is generally very interesting and holds significant value for the field.

*A: We understand and respect the evaluation of the reviewer. We thank the reviewer for all the relevant and valuable feedback. We hope that the answers to the reviewer's comments provided sufficient explanation. Furthermore, we hope that the changes we made to the manuscript substantially improved its quality and made it ready for a potential publication. Here is a summary of the major changes we made to address the reviewers' comments and improve the manuscript:*

- *We provided additional information on the selection of the temporal resolution of the meteorological datasets (chapter 2.3.2).*
- *We included an additional data set (Land Cover Change in 2050 - Global) to address future land cover changes within the future predictions. Accordingly, we updated all related figures and tables and provided new aspects to the discussion sections 4.2 and 4.4 and to the conclusions (5.).*
- *We provided partial dependence plots for the three most important predictors and the related forest fire susceptibility. We further provided partial dependence plots for the meteorological parameters (air temperature, precipitation) and the related forest fire susceptibility.*

● *We provided an additional figure showing the false positive rate (FPR) and false negative rate (FNR) as well as annual air temperature and precipitation averages for comparison. We briefly discussed this figure to analyse the model's mechanisms.*

*Further modifications:*

*We rephrased the sentence in 2.3.2 c) Anthropogenic influences & land use and land cover (LULC) (l. 172):*

*"Therefore, they **highly recommend** including…"*

*We rephrased the sentence in 4.4 Shortcomings and future perspectives (l. 446):*

*"Similarly, **forest fire products based on remote sensing data with a high spatial and temporal resolution**…"*

*We capitalized the word "**Supplement**" along the whole manuscript and Supplement.*

| Authors | Katharina Horn, Stenka Vulova, Hanyu Li, Birgit Kleinschmit |
|---|---|
| **Title** | Modelling Forest Fire Susceptibility in Brandenburg under Current and Future Scenarios |
| **Manuscript type** | Special Issue: Current and future water-related risks in the Berlin–Brandenburg region |
| **Journal** | Natural Hazards and Earth System Sciences (NHESS) |
| **Handling editor** | Axel Bronstert, axelbron@uni-potsdam.de |

*Editor comments are marked with "EC" and answers from the authors are marked with "A" in cursive. When applicable, we provided quotations of the modified sections from the manuscript to further illustrate the response. The modified sections that are quoted here are marked in **bold**.*

*We would like to warmly thank the editor for the extensive and helpful feedback. We hope that the integration of the reviewer's feedback has improved the quality of the manuscript and made it ready for publication.*
* * *
EC: Dear authors, thank you very much for your contribution. And thank you very much, for your revision of your manuscript according the suggestions of the reviewers. I think you followed most of their suggestions. I think this piece of work is rather relevant and should be published.

*A: Thank you very much for the positive feedback and the suggestion to publish our work. In the following, we will provide detailed answers to your comments.*

EC: However, I think one important aspects need to be more elaborated, for instance in chapter 4:
Choice , testing and validation of the climate scenarios:
• I miss a clear statement which regional climate scenarios were used, i.e. who provided these scenarios/ reference? What is their resolution? How about downscaling procedure?
• What is the reliability of these scenarios, in particular: Have they been validated how good they can represent current climate conditions? Is it only one realization or ensembles? Give a table with summarizes the most important values of the variables these scenarios (meteorological, land-use etc.)

*A: Thank you very much for pointing this out. It is true that we missed to address the details of the future climate scenarios within the manuscript so far. For the scope of this study, we used*

the Global Climate Model (GCM) MPI-ESM-1-2-HR. The climate data (monthly average minimum temperature (°C), monthly average maximum temperature (°C), and monthly total precipitation (mm)) were downloaded from WorldClim (www.worldclim.org). This website provides gridded multi-annual data sets based on different GCMs for different socio-economic pathways (SSPs) and different time periods between 2021 to 2100. WorldClim's downscaling procedure included comparing the projected weather data to the historical climate data and interpolating these changes to 30 arc seconds (~ 1 km) gridded data sets. The detailed downscaling procedure is described on the WorldClim website (https://www.worldclim.org/data/downscaling.html) (also see Fick and Hijmans, 2022). For further data processing, we computed the mean of the monthly average minimum and maximum temperatures (°C) for each grid cell.

*Reliability*

In terms of reliability of the GCM we considered the conclusions from Gutjahr et al. (2019) and Xu et al. (2023). In regard to the modeled air temperatures, Gutjahr et al. (2019) argue that these align closely to ERA-Interim, which is a climate reanalysis data set that has been previously used as a reference for assessing the quality of different GCMs (Xu et al., 2016, Gutjahr et al., 2019). Xu et al. (2023) assessed future drought characteristics of three CMIP6 models under SSP 5.85. Regarding air temperatures, the authors conclude that the MPI-ESM-1-2-HR scenario showed lower values than the other two scenarios (EC-Earth3, AWI-CM-1-1-MR). They further state that MPI-ESM1-2-HR along with AWI-CM-1-1-MR appeared rather reasonable in terms of estimating future drought conditions. As we consider future drought conditions vital to future forest fire susceptibility (FFS), we estimated that this GCM would suit our predictions best. Furthermore, the developers of the MPI-ESM-1-2-HR GCM argue that they aimed at the improvement of European climate simulations in particular (Gutjahr et al., 2019). Consequently, they focussed more closely on the effects of the North Atlantic as well as the Atlantic meridional overturning circulation. These aspects determined the decision of using future climate data from this GCM and determined the decision against using an ensemble of different GCMs.

The following table provides an overview of the data basis for the future scenarios. We added this table to the Supplement.

**Table S 7. Overview of the scenario's key variables used for the analysis of FFS in Brandenburg.**

| SSP | Time period | GCM | Land cover data | Mean air temperature (June 2081-2100) (°C) | Mean precipitation sum (June 2081-2100) (mm) |
|-----|-------------|-----|-----------------|--------------------------------------------|----------------------------------------------|
| 3.70 | June 2081-2100 | MPI-ESM- | "Current" land | 19.1 | 57.6 |

| 5.85 | | 1-2-HR (Gutjahr et al. 2019) | cover data (OpenStreetMap Contributors, 2023) | 20.4 | 53.7 |
|---|---|---|---|---|---|
| 5.85 * | | | Future land cover data (2050) (Esri Environment, 2021) | 20.4 | 53.7 |

*Accordingly, we added the following paragraph to the manuscript (ll. 85-93):*

*"The future scenarios of FFS cover the period of 2081 to 2100 using the socio-economic pathway (SSP) 5.85. SSPs are different projections of future greenhouse gas emissions under distinct potential political and socioeconomic developments. The SSPs range from SSP1.19 to SSP 5.85, covering CO2 concentrations ranging from 393 to 1135 ppm until 2100. SSP 5.85 represents "a high fossil-fuel development world throughout the 21st century" (Meinshausen et al., 2020).* **We decided to use SSP 5.85 from the Global Climate Model (GCM) MPI-ESM-1-2-HR. Xu et al. (2023) state that this GCM reflects future drought conditions rather well, which is why it was selected for this study. The climate data (monthly average minimum temperature (°C), monthly average maximum temperature (°C), and monthly total precipitation (mm)) were downloaded from WorldClim ([www.worldclim.org](www.worldclim.org)). This website provides gridded multi-annual data sets based on different GCMs for different socio-economic pathways (SSPs) and different time periods between 2021 to 2100 up to 30 arc seconds (~1 km) spatial resolution."**

*Additional references:*

*Gutjahr, O., Putrasahan, D., Lohmann, K., Jungclaus, J. H., Storch, J.-S., Brüggemann, N., Haak, H., & Stössel, A. (2019). Max Planck Institute Earth System Model (MPI-ESM1.2) for the High-Resolution Model Intercomparison Project (HighResMIP). Geoscientific Model Development, 12(7), 3241–3281. [https://doi.org/10.5194/gmd-12-3241-2019](https://doi.org/10.5194/gmd-12-3241-2019)*

*Xu, K., Yang, D., Yang, H., Li, Z., Qin, Y., & Shen, Y. (2016). Spatio-temporal variation of drought in China during 1961–2012: A climatic perspective. Journal of Hydrology, 526, 253–264. [https://doi.org/10.1016/j.jhydrol.2014.09.047](https://doi.org/10.1016/j.jhydrol.2014.09.047)*

EC: • I miss a discussion of the uncertainties of the climate variables for the future scenario periods. Regarding the meteorological scenarios, this is important particular for the rainfall values. The statement "precipitation is expected to increase in the future" is highly uncertain.

Therefore, the reader might also be interested in the sensitivity of this expected precipitation increase (i.e. what is the difference if a decrease in precipitation is assumed?).

*A: We thank the editor for addressing this. We agree that the statement needed to be elaborated further. Therefore, we added the following sentence to this section (ll. 348 to 355):*

*"Furthermore, meteorological projections assume that air temperatures will increase overall. However, precipitation is expected to increase in the future as well (see Fig. S 1 and S 2 in the Supplement). Consequently, wetter conditions may have lowered future FFS, outweighing the effect of higher air temperatures and contributing to the lower mean FFS in future scenarios compared to the extremely hot and dry year of 2022. **The German Weather Service (DWD, 2019) predicts changes between - 4% to + 13% in the annual precipitation sums until the end of the 21st century, illustrating the uncertainty of future precipitation predictions. As a result, in case of a decrease in precipitation until the end of the 21st century, this will strongly affect the flammability of Brandenburg's forests and thus the future FFS.** "*

*Additional reference:*

*DWD. (2019). Klimareport Brandenburg. Fakten bis zur Gegenwart—Erwartungen für die Zukunft (Report 1; p. 40).*
*https://www.dwd.de/DE/leistungen/klimareport_bb/klimareport_bb_2019_download.pdf;jsessionid=5BB70292853A51CB031B8D1E338E7F56.live21071?__blob=publicationFile&v=5*

EC: Furthermore, I think the conclusion could be a bit more reflect on open questions and uncertainties, e.g.:
• One of your main results is that distance to urban centres is most important. This requires a reflection of your definition of an "urban centre", and what kind of process you assume are behind such a rather crude variable (is it a higher intensity of car traffic? More hikers in the forests???)

*A: Thank you for pointing out the aspect of distance to urban settlements for predicting forest fire susceptibility. The data set we used to compute the distance to urban settlements (European Environment Agency [EEA], 2020b) shows impervious built-up areas at 10 m resolution. We assume that these areas are characterized by (ir-) regular human presence. As shown by several studies, anthropogenic influence plays a vital role in terms of forest fire ignition, which is why we consider this predictor key to reflect human presence.*

*Accordingly, we modified the following paragraph (ll. 177-181):*

*"Consequently, to include anthropogenic influences as well as aspects of LULC, distance to urban settlements, streets, railways, campsites, water bodies and military sites were selected as predictor variables. **According to the respective data set, we understand "distance to urban***

***settlements" as the distance to any type of constructed above-ground building (European Environment Agency [EEA], 2020b). We assume that this predictor can show (ir-) regular human presence at these places that may be related to an increased FFS."***

*Reference:*

*European Environment Agency [EEA]. (2020). Impervious Built-up 2018 (raster 10 m), Europe, 3-yearly, Aug. 2020. European Environment Agency.*

EC: What is the role of ammunition? You mention this only once in the beginning. Can this be reflected by the variable "Distance to a military sites"? Are FORMER military training sites includes in this variable?

*A: Thank you for addressing this. It is true that we did not elaborate on this any further throughout the manuscript. This predictor only includes both current and former military sites. As shown in Chapter 3, the distance to military sites was only moderately significant to the RF models (see Fig. 3). As we further show in Table S 6, the Wilcoxon test of the "Distance to military sites" predictor was not significant. Consequently, the distribution of the fire and non-fire classes did not significantly differ across Brandenburg. Therefore, we can conclude that the available data does not show a clear relationship between military sites (and ammunition) and an increased FFS.*

*Accordingly, we added the following sentences at the end of Chapter 4.1 (ll. 376 to 384):*

*"On a regional scale, anthropogenic parameters appear to be more relevant for FFS. In particular, the distance to urban settlements and railways showed a high significance for modelling FFS in Brandenburg. This confirms the statistics of forest fire emergence in Brandenburg provided by the Lower Forestry Authority of the State of Brandenburg (2023) (see Table S 2 in the Supplement), highlighting that most forest fires in Brandenburg emerge from human negligence or malicious arson. Several other studies have reached the same conclusion (Busico et al., 2019; Cilli et al., 2022; Ghorbanzadeh et al., 2019; Gnilke and Sanders, 2021; He et al., 2022; Ruffault and Mouillot, 2017).* ***However, the distance to military sites only moderately influenced the RF models (see Fig. 3). Furthermore, the Wilcoxon test (see Table S 6 in the Supplement) was not significant, underlining that there was no clear difference in the distribution of fire and non-fire points across Brandenburg. Therefore, the data and model results do not show a clear relationship between distance to military sites and FFS.***"

EC: What is the effect of the rather crude temporal resolution of one / three months? How much does this affect your results, and what kind of improvement would you expect for higher temporal resolution? You could elaborate on this in the concluding chapter.

*A: Thank you for your suggestion to elaborate on these aspects in the conclusion. We agree that incorporating these aspects enhances the quality of the conclusion.*

*We therefore added the following section to the Conclusion Chapter 5 (ll. 499 to 502):*

*"**The selection of a three-month temporal aggregation of the meteorological data sets was appropriate to reflect long-term meteorological trends. Using climate data at a higher temporal resolution would have shown the effect of extreme weather events more adequately. Therefore, future research could aim at integrating climate data at higher temporal resolution (e.g., weekly) to integrate the effect of extreme weather events into the predictions.**"*

We want to thank the referees for their final comments regarding the submitted manuscript. The referee comments (R1, R2) and authors' answers (A) can be found in the following. We have provided answers to the reports in *cursive*. Altered text passages that are cited in this document are marked in **bold**.

**Referee report #1**

R1: Thank you very much for resubmitting this very interesting manuscript and fully addressing my concerns. I think that including changes in land cover has greatly improved the manuscript. Well done! I have only one minor, rather technical comment. Other than that, I recommend this article for publication.

Minor comment:
- Lines: 348 – 351:

There it is written:

Furthermore, meteorological projections assume that air temperatures will increase overall. However, precipitation is expected to increase in the future as well (see Fig. S 1 and S 2 in the Supplement). Consequently, wetter conditions may have lowered future FFS, outweighing the effect of higher air temperatures and contributing to the lower mean FFS in future scenarios compared to the extremely hot and dry year of 2022.

I think here the wording regarding the wetter conditions is "too strong". In Fig. S1 and S2 you are technically comparing different data sources (DWD data and GCM data). So I think that concluding only from Fig. S1 and S2 that future precipitation is expected to increase is too strong. For instance, it could be also that MPI-ESM-1-2-HR is generally overestimating precipitation in June in this region. So the increase in precipitation could be mostly an effect of a general bias in the GCM compared to the station data from DWD. Although it becomes clearer that there are larger uncertainties in future precipitation patterns later, I would avoid this general statement here. Maybe it could help to rewrite those sentences clearly saying that this (only) refers to the meteorological input data that you are using.

*A: Thank you very much. We appreciate the positive feedback on the revision of the manuscript that have improved its quality. We understand your minor criticism regarding the word choice in this paragraph. It is true that the comparison of different data sources should be addressed more carefully here, especially considering the fact that we worked with a global climate model that may show local variances in future precipitation patterns. We thus modified this paragraph as follows:*

*"Furthermore, meteorological projections assume that air temperatures will increase overall. However, **the input data used for this study shows increased precipitation patterns in Brandenburg in the future scenarios compared to the periods of June 2016 and June 2022**  as well (see Fig. S 1 and S 2 in the Supplement). Consequently,  this change in precipitation patterns **shown by the input data** may have lowered future FFS **in the study region**, **thus** outweighing the effect of higher air temperatures and contributing to the lower mean FFS in future scenarios compared to the extremely hot and dry year of 2022."*

R2: The updated manuscript reflects thoughtful improvements, particularly in incorporating land-use change into the analysis. I'm pleased to see how this addition has provided deeper insights into a range of future results, even though it introduces an additional layer of complexity and potential limitations. The inclusion of more detailed metrics and trends for each predictor (e.g., with partial dependency plots) enhances the interpretability of the findings. The authors have also clarified the temporal aggregation methodology, now more transparently presented in the text.

*A: Thank you very much for the positive feedback on the updated manuscript. In the following, we will provide answers to your minor comments.*

Minor comments:

*Please consider publishing the FFS maps on Zenodo and the R code on GitHub, as providing access solely "on personal request" does not align with good scientific practice. I understand that the raw data might remain confidential, but the study's results and script should be accessible.

*A: Thank you for the suggestion. It is true that providing the codes and FFS maps on GitHub and Zenodo is beneficial for this publication. We have prepared both accordingly and added the information at the end of the manuscript under the section "Code availability":*

***"The code used for this study, as well as the forest fire susceptibility maps (Figs. 4 to 7) are publicly available on Zenodo at 10.5281/zenodo.14214918 and on GitHub at https://github.com/ka-horn/forest-fire-susceptibility-modelling."***

*Update "SSP 5.85" to "SSP5-8.5" as it represents Shared Socioeconomic Pathway 5 with a radiative forcing of 8.5 W/m² by the end of the 21st century.

*A: Thanks for pointing this out. We have updated all the occurrences of SSP scenarios in the text to "SSP1-1.9", "SSP3-7.0", and "SSP5-8.5", respectively. This also included updating the SSPs mentioned in Figures 2, 6 & 7.*

*Change the mean value for 2081–2100 in Table 3 to 0.417, as it still reflects the SSP3-7.0 value.

*A: Good catch, thank you very much for pointing this out. We corrected the mean value accordingly.*
* * *
*Additional comments from the authors:*
*We added the following sentence to the acknowledgement section: "**We acknowledge support by the Open Access Publication Fund of TU Berlin.**"*

*We added one word to the Figure caption of Figure 5: "Forest fire anomalies compared to 2016. Scenarios b and c both show predicted FFS **anomalies** in June 2081-2100 under SSP5-8.5. Scenario d includes projected land cover data, whereas scenario c does not. Border layer © 2018-2022 GADM."*